# Trends and Drivers of Soluble Iron Deposition from East Asian Dust to the Northwest Pacific: A Springtime Analysis (2001-2017)

Hanzheng Zhu[1,2,3], Yaman Liu[1,4], Man Yue[1,4], Shihui Feng[5], Pingqing Fu[5], Kan Huang[6], Xinyi Dong[1,2,3], Minghuai Wang[1,3]

[1] School of Atmospheric Science, Nanjing University, Nanjing, 210023, China
[2] Frontiers Science Center for Critical Earth Material Cycling, Nanjing University
[3] Joint International Research Laboratory of Atmospheric and Earth System Sciences & Institute for Climate and Global Change Research, Nanjing University, Nanjing, 210023, China
[4] Zhejiang Institute of Meteorological Sciences, Hangzhou, 310008, China
[5] Institute of Surface-Earth System Science, School of Earth System Science, Tianjin University, Tianjin, 300072, China
[6] Department of Environmental Science and Engineering, Fudan University, Shanghai, 200433, China

*Correspondence to*: Xinyi Dong (dongxy@nju.edu.cn)

**Abstract.** Recent shifts in dust emissions and atmospheric compositions in East Asia may have a significant impact on the deposition of soluble iron from dust over the Northwest Pacific. This study investigates the trends and driving factors behind this phenomenon during the springs of 2001-2017 using an enhanced version of the Community Atmosphere Model version 6 with comprehensive stratospheric chemistry (CAM6-chem). We improved the model to account for desert dust mineralogy and atmospheric chemical processes that promote iron dissolution, allowing for an in-depth analysis of the evolution of dust iron. Our findings indicate a decreasing trend in dust soluble iron deposition from East Asia to the Northwest Pacific by 2.4% per year, primarily due to reduced dust emissions driven by declining surface winds over dust source regions. Conversely, the solubility of dust iron showed an increasing trend, rising from 1.5% in 2001 to 1.7% in 2017. This increased iron solubility is linked to the acidification of coarse mode aerosols and in-cloud oxalate-promoted dissolution. Sensitivity model simulations reveal that the increase in anthropogenic $NO_x$ emissions, rather than the decrease in $SO_2$, plays a dominant role in enhancing dust aerosol acidity. This study highlights a dual trend: a decrease in the overall deposition of soluble iron from dust, but an increase in the solubility of the iron itself. It underscores the critical roles of both dust emission and atmospheric processing in promoting iron dissolution, which further influences soluble iron deposition and marine ecology.

## 1 Introduction

Dust aerosols profoundly influence the Earth's system (Kok et al., 2023) by absorbing and scattering radiation (Kok et al., 2017; Li et al., 2021), acting as cloud condensation nuclei (Karydis et al., 2017; Storelvmo, 2017), and providing essential nutrients to oceanic ecosystems (Mahowald, 2011). Among these impacts, the role of dust as a transporter of iron is particularly noteworthy due to its significant influence on marine productivity (Meskhidze et al., 2005; Jickells et al., 2005). The limited

availability of iron constrains biological productivity in vast expanses of the ocean particularly High Nutrient Low Chlorophyll (HNLC) regions (Martin, 1990; Martin et al., 1991). The deposition of iron-rich dust into the ocean acts as a much-needed nutrient source catalyzing the growth of phytoplankton (Hettiarachchi et al., 2021). The stimulation of phytoplankton growth not only boosts marine productivity but also has broader implications for carbon sequestration (Jickells et al., 2014). Understanding the dynamics of dust-borne iron deposition is critically important for comprehending and predicting changes in marine ecosystems and global climate (Jickells and Moore, 2015).

During atmospheric transport, atmospheric processing may significantly modify the bioavailability (solubility) of dust iron which is a critical factor for marine phytoplankton uptake. Observational studies have indicated that dust iron solubility could increase from as low as 2% to up to nearly 10% (Longo et al., 2016). The increase is primarily attributed to two atmospheric processes (Shi et al., 2012). The first involves interactions between dust aerosols and acidic gases or aerosols, leading to higher hydrogen ion concentrations that facilitate the dissolution of iron (Shi et al., 2011; Zhu et al., 2020). The second encompasses the formation of soluble iron-organic complexes, particularly through dust interactions with organic compounds like oxalate (Reichard et al., 2005; Paris and Desboeufs, 2013). Previous studies have simulated the atmospheric processing of dust iron by different models such as IMPACT (Ito and Xu, 2014; Ito, 2015; Ito and Feng, 2010), EC-Earth (Myriokefalitakis et al., 2022), GEOS-Chem (Johnson and Meskhidze, 2013) and CESM (Scanza et al., 2018; Hamilton et al., 2019) and indicated that these processes may play important role in altering the solubility of dust iron. However, the long-term trend of soluble iron due to the rapid changes in both natural dust emissions and anthropogenic emissions (Hamilton et al., 2020; Bergas-Massó et al., 2023) in recent decades remains poorly documented.

East Asia has two major dust emission sources: the Taklimakan and Gobi Deserts, which account for 14% of global dust emissions (Kok et al., 2023). It also has intensive anthropogenic emissions, which may result in increased iron solubility through the long-range transport of dust. Influenced by prevailing westerlies and monsoonal patterns, dust from East Asia could supply significant amounts of bioavailable iron to the Northwest Pacific and induce a substantial impact (Guo et al., 2017; Zan et al., 2023). Among the global HNLC regions which are the Subarctic Pacific, the Equatorial Pacific and the Southern Ocean, the Subarctic Pacific stands out due to its high sensitivity to iron fertilization (Martin and Fitzwater, 1988; Takeda and Tsuda, 2005). In recent decades, significant changes in natural mineral dust and anthropogenic emissions (Wu et al., 2022; Zheng et al., 2018) have necessitated a thorough evaluation of the contribution of East Asian dust to soluble iron deposition in the Northwest Pacific. Consequently, it is crucial to understand the trend and the driving factors to assess the impact accurately and develop informed environmental management strategies, and modeling is one of the best methods for this evaluation.

In this study, we aim to better understand the long-term trend and driving factors of dust soluble iron deposition from East Asia to the Northwest Pacific using an advanced model. We improved a global chemical-climate model to simulate the total and soluble dust iron emissions, transport, and deposition. Based on the simulation results, we quantified the interannual trend

in dust soluble iron deposition during spring from 2001 to 2017, considering contributions from both initial emissions and atmospheric processing which include proton-promoted and oxalate-promoted processes. Sensitivity experiments were conducted to explore the impact of anthropogenic emissions on dust soluble iron deposition in the Northwest Pacific. We also demonstrated a positive response from phytoplankton to dust soluble iron deposition by combining simulated high dust soluble iron depositions with satellite-derived sea surface chlorophyll concentrations.

## 2 Data and Methods

### 2.1 Model platform

Utilizing the Community Atmosphere Model version 6 with comprehensive stratospheric chemistry (CAM6-Chem), our study simulated the atmospheric dynamics of dust iron aerosols (Danabasoglu et al., 2020). Dust emissions are generated through the Dust Entrainment and Deposition model (DEAD), utilizing a geomorphic source function to accurately represent global soil erodibility variations (Zender et al., 2003a; Zender et al., 2003b). The model categorizes dust aerosols into three sizes: Aitken (0.01–0.1 µm), accumulation (0.1–1.0 µm), and coarse (1.0–10.0 µm), with specific geometric standard deviations at 1.6, 1.6, and 1.2 for each (Liu et al., 2016). Following the brittle fragmentation theory, the mass fractions of dust aerosol are configured to be 0.00165%, 1.1%, and 98.9% for each mode, respectively (Kok, 2011).

To refine atmospheric iron aerosol processing, we employed an updated CAM version coupled with the Model for Simulating Aerosol Interactions and Chemistry (MOSAIC) (Zaveri et al., 2021). In CAM6-Chem, SOA formation follows a Volatility Basis Set (VBS) approach with explicit VOCs and chemistry (Emmons et al., 2020; Tilmes et al., 2019). It incorporates wall-corrected SOA yields, photolytic removal of SOA, and more efficient removal by dry and wet deposition. What's more, the heterogeneous uptake of isoprene epoxydiols (IEPOX) onto sulfate aerosols and their subsequent production are explicitly simulated through coupling with MOSAIC (Jo et al., 2019; 2021). MOSAIC simulates the heterogeneous chemical processes on dust and is skilled at calculating size-resolved aerosol pH which has been widely used to explore aerosols' impact on air quality, climate, and health (Ruan et al., 2022; Liu et al., 2023). Firstly, the MOSAIC mechanism would determine whether the aerosol contains solid $CaCO_3$ which can adsorb acidic gases ($H_2SO_4$, $HNO_3$, and $HCl$). The irreversible heterogeneous reactions which would consume solid $CaCO_3$ have been listed in Table S1. Before $CaCO_3$ was consumed, the aerosol pH was dependent on the hydrogen ions dissociated from water instead of acids (Zaveri et al., 2008). This process was influenced by temperature (T) of which the unit is K as follows (Eq.1):

$$pH = \sqrt{1.01^{-14}e^{-22.52\left(\frac{1}{T}-1\right)+26.92\left(1+log_{10}\left(\frac{1}{T}\right)-\frac{1}{T}\right)}} \tag{1}$$

After $CaCO_3$ was consumed, the MOSAIC defined two domains (sulfate-rich and sulfate-poor) to determine aerosol pH as follows (Eq.2):

$$X_T = \frac{m_{Na^+} + m_{NH_4^+} + 2m_{Ca^{2+}}}{m_{SO_4^{2-}} + m_{HSO_4^-}} \tag{2}$$

where m is the concentration of ions in aerosol water and the unit is mol/kg.

Under sulfate-rich conditions ($X_T < 2$), the aerosol tends to absorb negligible HNO$_3$ and HCl due to the high acidity and sulfate have a pronounced effect on aerosol pH as follows (Eq.3):

$$m_{H^+} = 2m_{SO_4^{2-}} + m_{HSO_4^-} + m_{NO_3^-} + m_{Cl^-} - (2m_{Ca^{2+}} + m_{NH_4^+} + m_{Na^+}) \tag{3a}$$

$$K_{HSO_4^-} = (\frac{m_{H^+}m_{SO_4^{2-}}}{m_{HSO_4^-}})(\frac{\gamma_{SO_4^{2-}}\gamma_{H^+}}{\gamma_{HSO_4^-}}) \tag{3b}$$

where $K_{HSO_4^-}$ is the equilibrium constant of the bisulfate ion dissociation and the unit is mol$^2$ kg$^{-2}$ atm$^{-1}$ and $\gamma$ is the activity coefficient of electrolyte calculated by MOSAIC (Zaveri et al., 2005a).

Under sulfate-poor conditions ($X_T > 2$), it is important to account for the gas-aerosol exchange of semi-volatile gases such as HNO$_3$, HCl, and NH$_3$. Using the internal equilibrium H+ concentration would lead to oscillations in the condensation and evaporation of these gases, as it does not provide steady-state results (Zaveri et al., 2008). Therefore, MOSAIC employs

dynamic H+ concentrations which is determined by gas-particle exchange of semi-volatile acidic gases (HNO$_3$ and HCl) predominantly controls aerosol acidic pH as follows (Eq.4):

$$m_{H^+} = \frac{K_{HNO_3}^{gl} C_{l,HNO_3}}{\kappa_{HNO_3} m_{NO_3^-} (\gamma_{HNO_3})^2}, or \tag{4a}$$

$$m_{H^+} = \frac{K_{HCl}^{gl} C_{l,HCl}}{\kappa_{HCl} m_{Cl^-} (\gamma_{HCl})^2} \tag{4b}$$

where m is the concentration of ions in aerosol water, K is the gas-particle equilibrium constant of acidic gases, $C_l$ is the

equilibrium concentrations of acidic matter for the liquid phase and the unit is mol m$^{-3}$ (air), $\kappa$ is the first-order mass transfer coefficient and $\gamma$ is the activity coefficient of electrolyte calculated by MOSAIC (Zaveri et al., 2008; Zaveri et al., 2005a; Zaveri et al., 2005b). Specifically, equation (4a) is applied when HNO$_3$ gas and NO$_3$ aerosol are both present and have concentrations greater than zero while Equation (4b) is used when HCl gas and chloride aerosol concentrations are greater than zero after computing equilibrium surface concentrations. And the aerosol pH is calculated based on H$^+$ concentrations for

each aerosol mode at each time step.

The standard official version of CAM6-Chem doesn't simulate dust mineralogy or the atmospheric processes of iron solubility change. Instead, it assigns a total iron fraction of 3.5% to dust and the dust iron deposition solubility relies on the ratio of coarser dust to fine dust fluxes (Long et al., 2021). In order to properly simulate the diversity of dust iron emissions from different deserts, we developed the model to incorporate five main iron-contained minerals. Each of these minerals includes three distinct dust iron types, thereby providing a more accurate representation of dust mineralogy. We also developed pH-promoted and oxalate-promoted processing in the model to simulate dust iron solubility changes by following the parameterization schemes of Hamilton et al. (2019). More details of model development conducted in this study are described in the next subsection.

## 2.2 Model development of dust mineralogy and atmospheric processing

In this study, a detailed mineralogy map database (Nickovic et al. 2012) was implemented into the CAM6-chem model to configure the mineral composition of dust emission. This map is based on the work of Claquin et al. (1999) and has been widely used in previous modeling studies (Johnson and Meskhidze, 2013; Ito and Xu, 2014; Myriokefalitakis et al., 2015). The map segments soil into silt/clay fractions of different minerals which include five main iron minerals (hematite, smectite, illite, kaolinite, and feldspar). Notably, compared to the mineralogy map from Journet et al. (2014), this mineralogy map has been demonstrated to have a good performance, especially in identifying phyllosilicates with high soluble iron content (Gonçalves Ageitos et al., 2023). The fractions of five iron minerals in dust aerosols across three modes were determined based on mineral soil distributions (Scanza et al., 2015). Implementing the mineralogy map allows the model to describe dust iron emission from different deserts other than using a uniform fixed fraction of iron content, which is important for characterizing regional diversity. On top of the mineralogy, we further grouped dust iron into three solubility types—Slow-soluble ($Fe\_ss$), Medium-soluble ($Fe\_ms$), and Readily-soluble ($Fe\_rs$)—with proportions aligned with Hamilton et al. (2019) and Scanza et al. (2018). The detailed iron content and initial iron solubility for each of the five minerals have been shown in Table S2. These values are sourced from measurements (Journet et al., 2008; Shi et al., 2011a; Shi et al., 2011b) and are consistent with previous modeling studies (Ito and Xu 2014; Scanza et al., 2018; Hamilton et al., 2019). Using the mineralogy map, our model achieved to simulate the global spatial patterns of total and initial soluble iron emissions. The total iron content in dust aerosol is higher in the main dust sources including North Africa, Middle East and central Asia, and East Asia, is higher than the default setting of 3.5% (Fig. S1a). This is consistent with the observations (Lafon et al., 2004, 2006; Shi et al., 2011b) and the research by Ito and Xu (2014), which reported that the observed dust iron content in North Africa and East Asia averaged 3.7%. Therefore, the use of the mineralogy map increases the iron content in dust from these regions (Fig. S1b) that suggest the default settings likely underestimate dust iron in these main dust source regions.

We also developed two atmospheric processing of dust iron aerosols via proton-promoted and oxalate-promoted mechanisms in the model. For proton-promoted dissolution, we employ the first-order dissolution rate formula of Lesaga et al. (1994) as follows (Eq.5) for interstitial iron aerosols:

$$RFe_{i,proton} = K_i(T) \times a(H^+)^{m_i} \times f(\nabla G_r) \times A_i \times MW_i \tag{5a}$$

$$\frac{d}{dt}[Fe_{sol,\ proton}] = RFe_{i,proton} \times [Fe_{insol}] \tag{5b}$$

where $RFe_{i,proton}$ is the proton-promoted iron dissolution rate of which unity is s$^{-1}$ and i represents two types of iron (Fe_ss and Fe_ms), $K_i(T)$ is the temperature-dependent rate coefficient (Meskhidze et al., 2005), $a(H^+)^{m_i}$ is the proton concentration with an empirical reaction order $m_i$, $f(\nabla G_r)$ accounts for the change in the dissolution rate with variation from equilibrium which equals 1 for simplicity, A is the specific surface area and MW is the molecular weight. The parameters for the first-order dissolution rate formula (Eq. 5a) are based on previous studies (Meskhidze et al., 2003; Ito and Feng, 2010; Ito and Xu, 2014) and are aligned with Scanza et al. (2018) and Hamilton et al. (2019). For $K_i(T)$ in Fe_ms type, here we use the dissolution rate of mineral illite as an additional simplification following Scanza et al., (2018). In contrast, some studies (Ito and Feng, 2010; Ito, 2012; Ito and Xu, 2014) employ separate dissolution rates for different minerals. For $K_i(T)$ in Fe_ss type, we use a fast dissolution rate with three stages, following Ito and Xu (2014). The proton-promoted iron dissolution rate is influenced by factors including temperature, dust iron type, and aerosol pH. The simulated aerosol pH in accumulation and coarse mode are shown in Figure S2. The fine particles are relatively more acidic while coarse-mode particles are significantly less acidic influenced by sea salt and dust components. The comparison of annually averaged accumulation mode aerosols' pH with observations collected by Pye et al. (2020) shows that, our model successfully captured the global characteristics of fine aerosol pH. The correlation coefficient and normalized mean bias (NMB) are 0.4 and 27% respectively. The discrepancy could be attributed to the seasonal variations and the dynamics of precursor gas emissions, environmental factors such as relative

humidity.

For oxalate-promoted dissolution, we employ the first-order dissolution rate formula for cloud-borne iron aerosols as follows (Eq.6):

$$RFe_{i,oxalate} = a_i \times [C_2O_4^{2-}] + b_i \tag{6a}$$

$$\frac{d}{dt}[Fe_{sol,\ oxalate}] = RFe_{i,oxalate} \times [Fe_{insol}] \tag{6b}$$

where $RFe_{i,oxalate}$ is the oxalate-promoted iron dissolution rate of which unity is s$^{-1}$ and i represents two types of iron (Fe_ss and Fe_ms), coefficients $a_i$ and $b_i$ are determined by Paris et al., (2011) and aligned with Scanza et al., (2018) and Hamilton et al., (2019). This linear relationship between oxalate-promoted dissolution rate and oxalate concentration in solution is based on cloud water studies by Paris et al. (2011) and has been employed in previous modeling studies (Johnson and Meskhidze, 2013; Myriokefalitakis et al., 2015). Furthermore, Ito and Shi (2016) developed a new oxalate-promoted scheme, which has

been applied in recent research (Myriokefalitakis et al., 2022). The default CAM6-Chem did not include an explicit chemistry of oxalate. Therefore, we employ the formula from Hamilton et al., (2019) as follows (Eq.7) to estimate oxalate concentrations:

$$[C_2O_4^{2-}]_{lon,lat,lev} = 150 \times \frac{[SOA]_{lon,lat,lev}}{\max[SOA]} \tag{7}$$

where $[C_2O_4^{2-}]_{lon,lat,lev}$ is the estimated oxalate concentration and $[SOA]_{lon,lat,lev}$ is the modeled secondary organic aerosol concentration. The formula has been demonstrated to estimate oxalate concentration accurately as both oxalate and SOA are

the product of the oxidation of volatile organic carbon gases (Hamilton et al, 2019; Myriokefalitakis et al., 2011). The upper threshold of oxalate concentration is 15 μmol L$^{-1}$ keeping consistent with Hamilton et al. (2019) and Scanza et al. (2018). Because the SOA burden simulated in our model version (Figure S3) is comparable with the previous version (Fig. 5a; Liu et al., 2012). The maximum SOA concentration was similar to the study of Hamilton et al. (2019). For the oxalate evaluation, we have collected global oxalate observations in rain/cloud water to evaluate our model results. The locations and months are

consistent between observations and the model. We utilized monthly mean modelled values, averaged climatologically over the period 2001-2017. Comparisons with observed oxalate levels indicate that our model accurately captures the quantitative characteristics of oxalate especially in East Asia (Figure S4). Furthermore, we tag soluble iron into two types: Fe_ps representing those produced from the proton-promoted processing, and Fe_os representing those produced from the oxalate-promoted processing, enabling an evaluation of these processes' contributions during atmospheric transport.

In this study, simulations of 17 springs from 2001 to 2017 were conducted to analyze long-term spatiotemporal characteristics of dust iron (Table 1). By analyzing the simulation results for the whole year, we found that the spring accounted for 60% of dust deposition and 55% of dust soluble iron deposition over the Northwest Pacific of a full year (Figure S5). Hence, we concentrate our simulation analysis on spring only in this study. For model evaluation purposes, we conducted simulations that aligned with the periods when North Pacific iron observations were available. We also conducted a full-year simulation

in 2013 which aligns with the most observations' periods for model evaluation. Two sensitivity experiments were conducted to explore the impact of anthropogenic emissions. All simulations maintained a horizontal resolution of 0.95° by 1.25° (latitude by longitude) and a vertical resolution extending to approximately 40 km with 32 layers (Emmons et al., 2020). Meteorological data from the Modern-Era Retrospective analysis for Research and Applications (MERRA2) reanalysis dataset (Gelaro et al., 2017) were utilized for model adjustment by offline nudging with a relaxation time of 1 hour, minimizing meteorological

uncertainty. Anthropogenic and biomass burning emissions follow CMIP6 protocols up to 2014 (Eyring et al., 2016), transitioning to the SSP585 scenario thereafter (O'Neill et al., 2017), with emissions over China refined using the MEIC inventory for enhanced local accuracy (Li et al., 2017a; Zheng et al., 2018; Yue et al., 2023).

**Table 1. Sensitivity experiments' configurations**

| Experiments | Period | Anthropogenic SO$_2$ emissions | Anthropogenic NO$_x$ emissions |
|---|---|---|---|
| Base | 2001-2017 spring | | |
| Iron evaluation | Align with observations | Updated emissions from MEIC | |
| pH evaluation | 2013 | | |
| SO$_2$ change | 2001 spring | 2007 spring | 2001 spring |
| NO$_x$ change | 2001 spring | 2001 spring | 2007 spring |

**2.3 Observations**

The study employs Dust Optical Depth (DOD) at 550nm from the MODIS Dust Aerosol (MIDAS) dataset which has a fine resolution of 0.1°×0.1° and contains daily DOD over 2003-2017 (Gkikas et al., 2021). Hourly PM$_{10}$ observational data from 1571 sites across China over 2015-2017 springs are sourced from the China National Environmental Monitoring Centre (CNEMC; http://www.cnemc.cn/). The 230 observational records (~40% are spring) of total and soluble iron concentrations in the North Pacific over 2001-2018 are from the reported articles (Mahowald et al., 2009; Myriokefalitakis et al., 2018) and GEOTRACES Intermediate Data Product Group (IDP2021v2). Additionally, daily chlorophyll-a concentrations are obtained from the Ocean-Colour Climate Change Initiative (OC-CCI, v6.0) products (Sathyendranath et al., 2019).

## 3 Results and Discussion

### 3.1 Model evaluation

#### 3.1.1 Dust surface concentrations and observed PM$_{10}$

We first compared simulated dust surface concentrations against observed PM$_{10}$ levels in Eastern Asia (EA) over 2015-2017 to evaluate model performance in representing dust aerosol mass concentration (Figure 1). Observed PM$_{10}$ is a recognized proxy for dust concentrations during high dust event frequencies (Wang et al., 2021; Li et al., 2024). The comparison confirmed a strong concordance between the modelled spatial distribution of dust and observed PM$_{10}$ values, underscoring the reliability of our model. Dust emitted from the Taklimakan and Gobi deserts was mainly transported eastward and southward, impacting the Northwest Pacific (NWP: 30-50N, 140E-160W). Here we defined the Eastern Sources region (EAS: 30-50N, 90-130E) to represent the origins of dust that may subsequently transport to and deposit over the NWP.

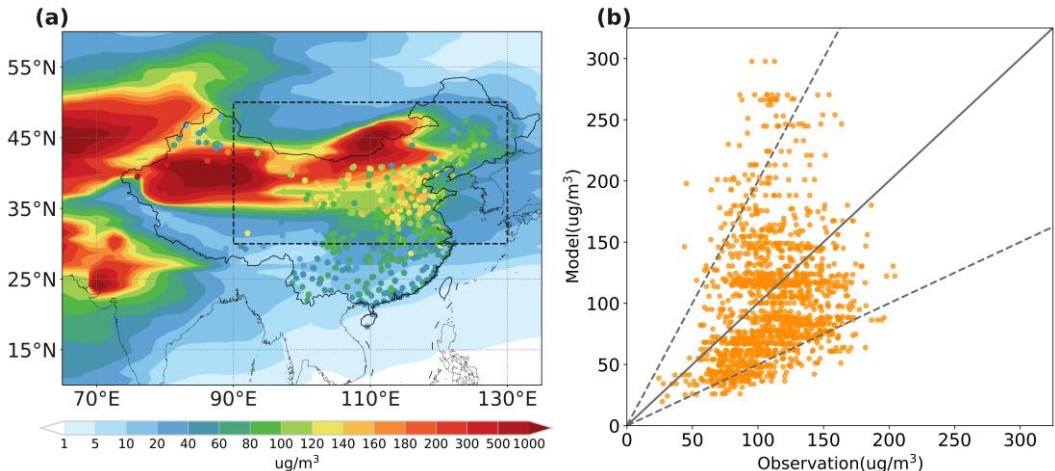

**Figure 1: (a) Simulated dust surface mass concentrations and observed PM$_{10}$ (dots) on average of 2015-2017 springs. (b) A linear relationship between spring averaged simulated dust surface mass concentrations and observed PM$_{10}$ over Eastern Sources (EAS: 30-50N, 90-130E) during 2015-2017 springs, the solid black line represents y=x, the dotted black lines represent y=0.5/2x.**

Specifically, evaluation against ground-based measurements suggested a good consistency between simulated dust concentrations and observed PM$_{10}$ data over the EAS (Fig. 1b). The average simulated dust surface mass concentration (104 µg/m³) closely mirroring the observed PM$_{10}$ concentration (110 µg/m³) over the EAS. The error margin for the simulated dust concentrations varies between half and twice that of the observed values, indicated by dotted black lines in Fig. 1(b). And the normalized mean bias (NMB) is approximately -5.5%. Our model tends to overestimate dust concentrations in the eastern Gobi Desert to some extent. In the North China Plain, the simulated concentrations are slightly lower than the observed PM$_{10}$ due to high anthropogenic emissions. Overall, these comparisons confirm the proficiency of our model in accurately replicating dust mass concentrations as compared with other related studies (Wang et al., 2021; Liang et al., 2022).

### 3.1.2 Dust burden and MODIS dust optical depth

The averaged dust burden in the EAS and NWP regions for the springs of 2001-2017 were 2.7 and 0.3 Gg respectively with the spatial distribution shown in Fig. 2(a). Dust particles deposited rapidly so the burden also gradually decreased from near desert areas to remote ocean during the long-range transport. With respect to the long-term trend, it is well-acknowledged that dust over East Asia significantly declined over 2001-2017 (Wu et al., 2022). The MODIS DOD product also suggested a decreasing trend by -24% and -27% per decade over EAS and NWP respectively as shown in Fig. 2(b) and Fig. 2(c). Our model results generally agreed well with other studies in reproducing this decreasing trend. The simulated dust burden trend was -3.4% and -8.3% per decade over EAS and NWP respectively. Although the interannual variations between the model and MODIS were different for some specific years and our simulation underestimated the magnitude of the declining trend, the model was demonstrated to successfully reproduce the long-term decreased trend of dust over both land and ocean.

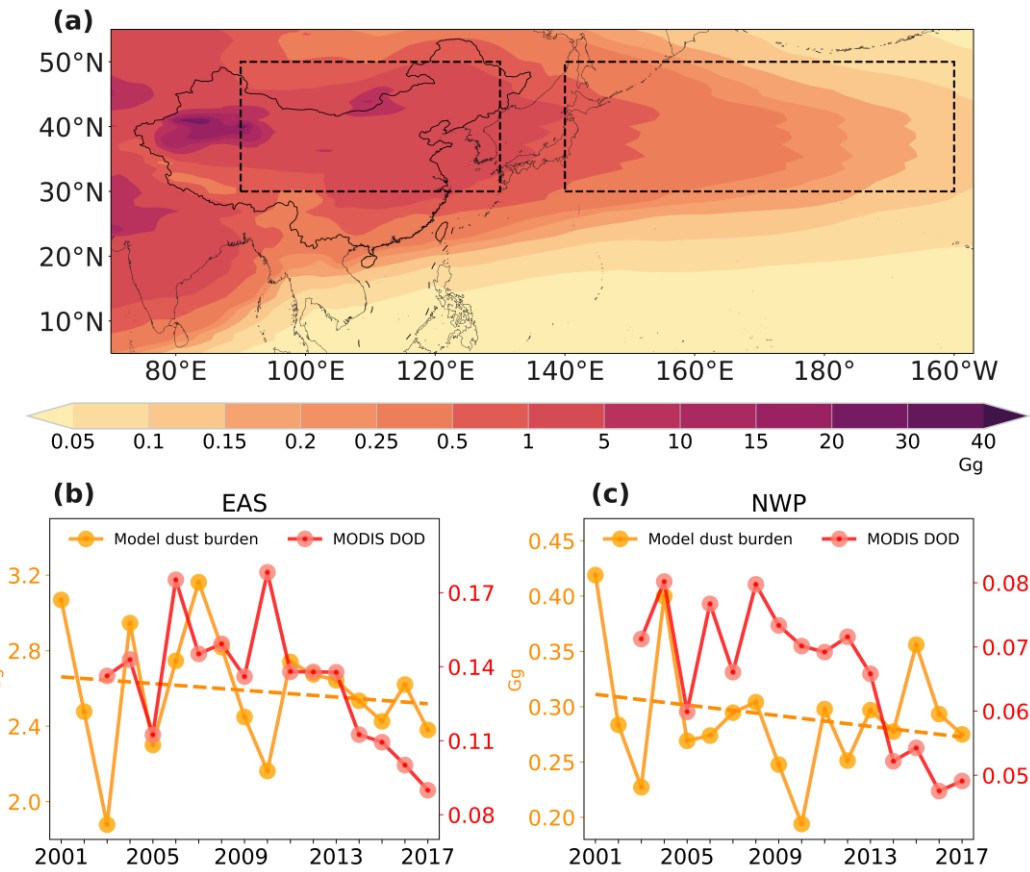

**Figure 2: Spatial distributions of simulated dust burden (a) on average of 2001-2017 springs. Temporal variations of spring averaged simulated dust burden (yellow line) and MODIS DOD (red line) over (b) Eastern Sources (EAS: 30-50N, 90-130E) and (c) Northwest Pacific (NWP: 30-50N, 140E-160W) during 2001-2017 springs.**

### 3.1.3 Total and soluble iron concentrations

We compared model outputs with observed data for total and soluble iron concentrations over the North Pacific as shown in Figure 3. The sampling times and locations of the observed data are presented in Figure S6. The evaluation ensured temporal
and spatial alignment of the simulated data with these observations. Influenced predominantly by dust activities, both total and soluble iron concentrations are generally higher in spring and lower in winter, as per the observational data. Our model captured these seasonal variations, reinforcing the hypothesis that dust is a major factor influencing iron concentrations in the North Pacific.

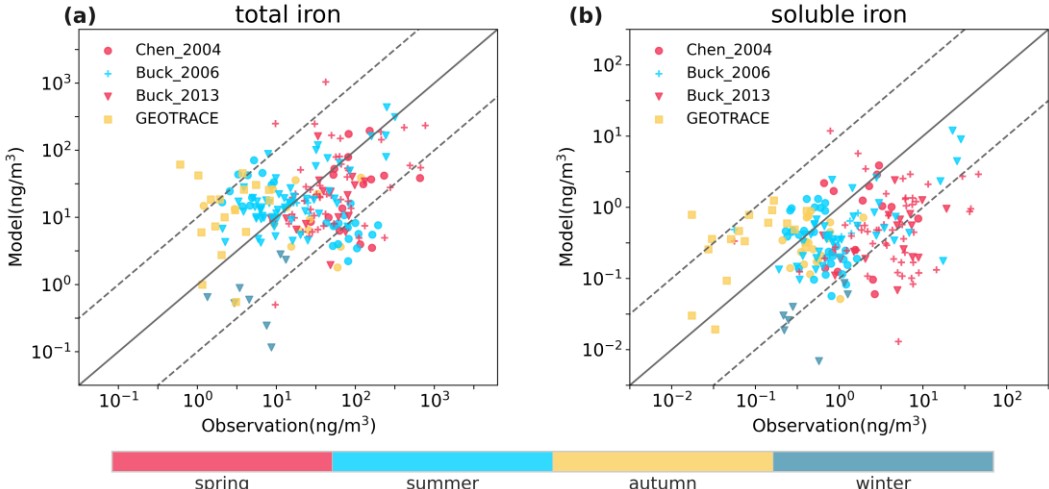

**Figure 3: A linear relationship between simulated total (a) and soluble (b) iron concentrations and cruise observations (Chen, 2004; Buck et al., 2006; Buck et al., 2013; GEOTRACES Intermediate Data Product Group (2021)). Different colors and shapes present observations' seasons and cruises respectively.**

The agreement between the model's predictions and ship-based observed data for iron concentrations evidenced our model's ability to simulate iron aerosols. We also compare the performance between the default CAM6-chem model and our improved
model in terms of validation against observations. The developed model achieved a reduction in NMB from -82% to -77% for total iron and from -44% to -33% for soluble iron. For total and soluble iron concentrations, the Pearson correlation coefficients (r) between the simulated from developed model and observations are 0.27 and 0.36. Our developed model's evaluations are comparable with other model results (Myriokefalitakis et al., 2022) and it demonstrates our model's good ability to capture the characteristics of total/soluble iron over the North Pacific.

The observations of iron used here are the total iron which includes dust and pyrogenic iron. The simulated results are lower than observations likely due to the lack of pyrogenic iron. What's more, the comparison about iron solubility between

simulation and observations is shown in Figure S7. Our model only calculates iron solubility between 0 and 10%. This is likely due to the lack of pyrogenic iron which has been suggested to contribute to higher iron solubility (Ito et al., 2019).

## 3.2 Spatial and temporal characteristics of dust iron deposition

**3.2.1 Atmospheric dust iron budget**

The global mean emissions of dust, dust total iron, and dust soluble iron were 2707 Tg/yr, 109 Tg/yr, and 0.98 Tg/yr, respectively, based on our 2017 model simulation (Table S3). These values are comparable to those reported by the Mechanism of Intermediate complexity for Modelling Iron (MIMI) model (Hamilton et al., 2019) (3200 Tg/yr and 130 Tg/yr for dust and dust total iron emissions) but are approximately twice as high as the results from EC-Earth model (Myriokefalitakis et al., 280 2022) (1265 Tg/yr and 59.3 Tg/yr). The simulated global mean iron content in dust is 4.0% which aligns well with MIMI (4.1%) and EC-Earth (4.7%). The initial iron solubility is 0.91% and higher than the 0.1% set by EC-Earth.

The global annual atmospheric dust total and soluble iron budget have been presented in Table 2. The simulated global burdens of dust total and soluble iron were 1331 Gg and 15.4 Gg respectively. The simulated global dust total and soluble iron deposition were 109 and 1.3 Tg/yr aligning well with values reported by the MIMI (Table 3). Due to higher dust emissions, 285 our simulated dust iron deposition and burdens are nearly double those of the EC-Earth and Ensemble models (Myriokefalitakis et al., 2018). For solubilization rates, the EC-Earth model reported 315 Gg/yr and 170 Gg/yr for proton-promoted and oxalate-promoted dissolution but our results indicated 129 Gg/yr and 200 Gg/yr. The lower solubilization rate of proton-promoted and higher solubilization rate of oxalate-promoted in our study could be attributed to the lower simulated coarse aerosol mode acidity and higher scaled oxalate levels than EC-Earth respectively. And the difference in simulation period and iron 290 dissolution mechanisms would also induce discrepancy. For the Northwest Pacific, simulated dust total and soluble iron burdens were 7.4 Gg and 0.11 Gg aligning closely with Ensemble model results. Dust total and soluble iron deposition rates in this region were 589 Gg/yr and 10.1 Gg/yr consistent with both MIMI and Ensemble model estimates. But EC-Earth reported higher NWP deposition rates than our model due to differences in regional dust simulation.

**Table 2. Global annual atmospheric dust total/soluble iron budget in 2017.**

| | Burden (Gg) | | Dry deposition (Tg/yr and Gg/yr) | | Wet deposition (Tg/yr and Gg/yr) | | Solu. Rate (Gg/yr) | |
|---|---|---|---|---|---|---|---|---|
| | Fe_tot | Fe_sol | Fe_tot | Fe_sol | Fe_tot | Fe_sol | Fe_ps | Feos |
| **Global** | 1331 | 15.4 | 42.7 | 445 | 66.5 | 874 | 129 | 200 |
| **coarse, fine** | 1301, 30 | 13.3, 2.1 | 42.2, 0.5 | 421, 24 | 65.4, 1.1 | 789, 85 | 78, 51 | 196, 4 |
| **NWP** | 7.4 | 0.11 | 0.05 | 1.2 | 0.54 | 8.9 | 1.3 | 1.73 |
| **coarse, fine** | 7.1, 0.3 | 0.09, 0.02 | 0.05, 0.001 | 1.1, 0.1 | 0.53, 0.01 | 7.8, 1.1 | 1.1, 0.2 | 1.68, 0.05 |

**Table 3. Comparison of global annual atmospheric dust total/soluble iron budget from different studies.**

| | | This study | | MIMI[1] | | EC-Earth[1] | | Ensemble[1] | |
|---|---|---|---|---|---|---|---|---|---|
| | | Fe_tot | Fe_sol | Fe_tot | Fe_sol | Fe_tot | Fe_sol | Fe_tot | Fe_sol |
| **Global** | **Burden (Gg)** | 1331 | 15.4 | / | / | / | 6 | 563 | 10 |
| | **Deposition (Tg/yr)** | 109 | 1.3 | 127 | 1.6 | 59 | 0.6 | 68 | 0.6 |
| **NWP** | **Burden (Gg)** | 7.4 | 0.11 | / | / | / | / | 5.7 | 0.18 |
| | **Deposition (Gg/yr)** | 589 | 10.1 | 472 | 10.4 | 2147 | 33 | 369 | 7.7 |

[1.] *The simulation periods are as follows: 2007-2011 for MIMI, 2000-2014 for EC-Earth, and 2007-2014 for the Ensemble.*

We find through the simulation that dust iron deposition and iron solubility show prominently different spatial distribution patterns (Fig. 4a and Fig. 4c). Dust iron deposition gradually decreased from the source region to the remote ocean, whereas dust iron solubility gradually increased along the transport pathway. In the mainland of EA, simulated dust iron solubility ranged from 0 to 1.4%, aligning with observations (Ooki et al., 2009; Shi et al., 2020; Zhang et al., 2023a). Over the downwind oceanic areas, elevated dust iron solubility levels were found especially in the remote NWP (30-50N, 160E-160W; marked in

Figure 4) which could be up to 2%. The increasing trend of iron solubility from land to ocean implied the significant influence of atmospheric processing in altering dust iron during long-range transport.

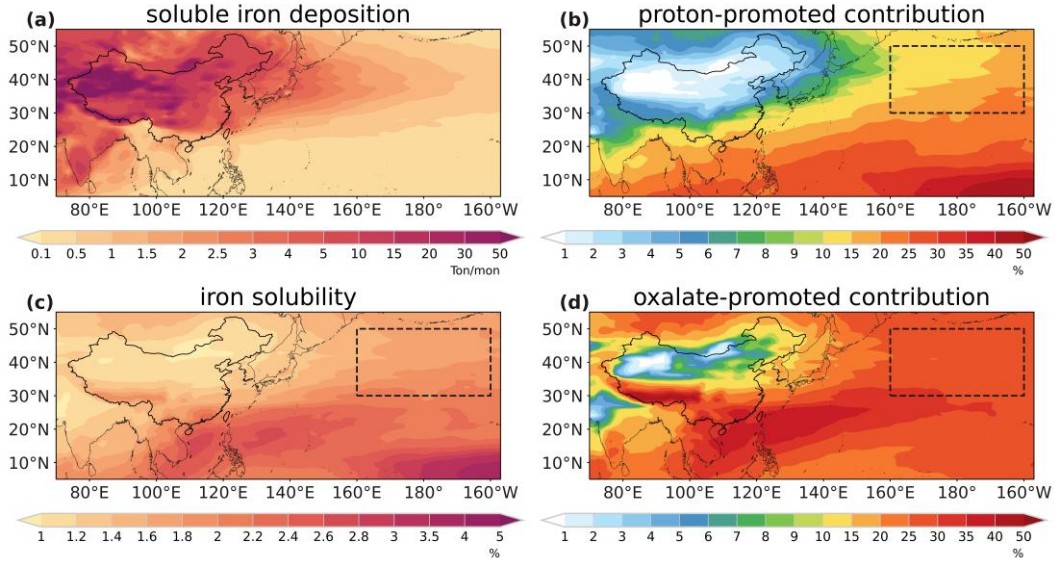

**Figure 4: Spatial distributions of (a) soluble iron deposition, (c) iron solubility, (b) proton-promoted relative contributions to soluble iron deposition, and (d) oxalate-promoted contributions to soluble iron deposition on average of 2001-2017 springs.**

We further investigated the contributions of proton-promoted and oxalate-promoted processing to soluble iron deposition (Fig. 4b and Fig. 4d). Over the Taklimakan and Gobi Deserts, atmospheric processing contributions show little influences on iron solubility due to lack of acids and oxalate. However, their contributions significantly increased with distance from land along

the transport pathway, reaching nearly 40% over the remote NWP. This underscores the important role of atmospheric processing in dust iron deposition and dust iron solubility especially over remote ocean areas.

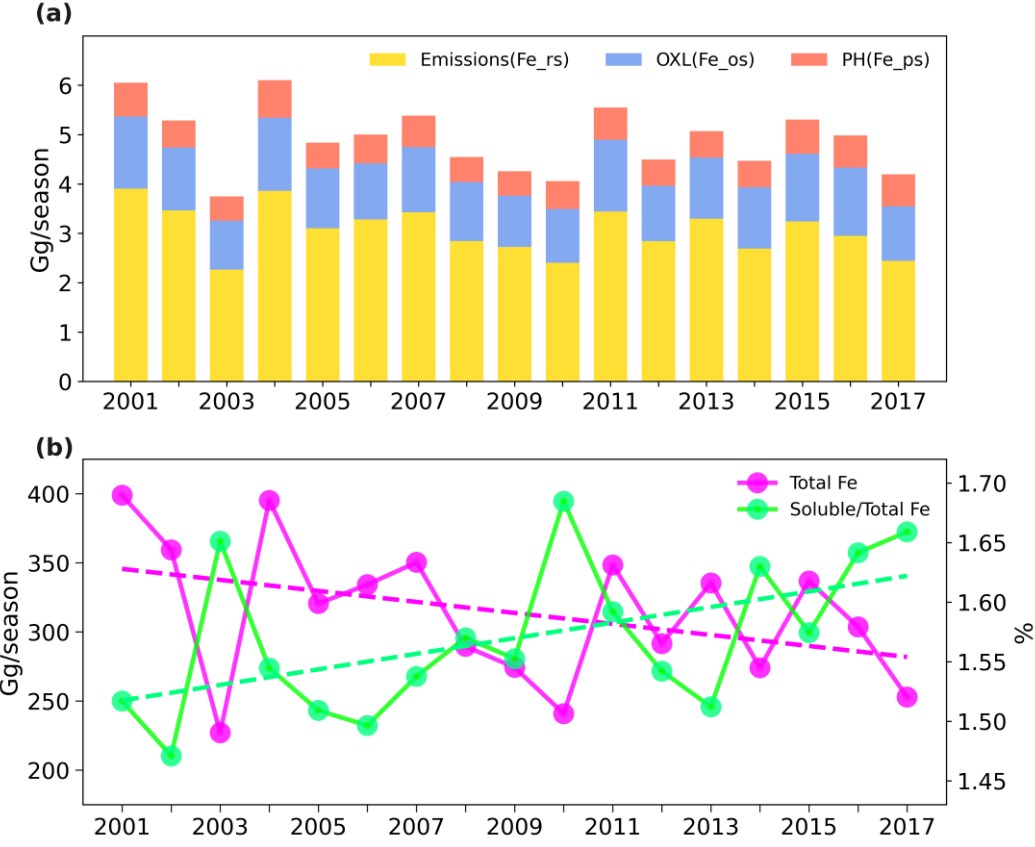

**Figure 5: (a) Temporal variations of total soluble iron deposition (stacked bar) and separated soluble iron deposition from emissions (yellow bar), proton-promoted processing (red bar), and oxalate-promoted processing (blue bar) during 2001-2017 springs. (c) Temporal variations of total iron deposition (green solid line) and the ratio of soluble to total iron deposition (purple solid line) during 2001-2017 springs.**

Throughout the springs of 2001-2017, the NWP received an average of 4.9 Gg/season of soluble iron deposition from EA (Figure 5). The relative contributions of emissions, proton-promoted and oxalate-promoted to Northwest Pacific dust soluble iron deposition in coarse and fine modes are presented in Figure S8. Atmospheric processing played a significant role (~40%) in dust soluble iron deposition of which the oxalate-promoted processing emerged as a dominant contributor (25%). The contribution of the oxalate-promoted processing was about twice that of proton-promoted processing in the NWP. This finding

is consistent with global results (Table 2) and aligns with previous global modelling (Johnson and Meskhidze, 2013; Scanza et al., 2018) and East Asian observational research (Shi et al., 2022). Differently, Ito and Shi (2016) and Myriokefalitakis et al. (2022) found that proton-promoted dissolution is the primary process. The higher contribution of oxalate-promoted dissolution in our study might be partially attributed to differences in the parameterization of oxalate concentrations and aerosol

acidity between models. As the oxalate concentrations appear to be underestimated in their model and the simulated mainland coarse-mode aerosol acidity in our model (Figure S2) is obviously lower than those. Furthermore, regional differences can also play a role, as our model performs well for oxalate concentrations over East Asia but not elsewhere (Figure S4). Future studies should focus on improving these aspects to refine the relative contributions of atmospheric processing. As for different size aerosols' contributions, our modelling results underscore the dominant role of coarse model aerosols which accounted for nearly 90% dust soluble iron deposition in the NWP. Fine mode (Aitken + accumulation) aerosols have a higher iron solubility and could travel a longer distance, but the relatively low mass concentration limited the contribution to iron deposition. And the dominant role of oxalate-promoted processing was mainly determined by the coarse mode (Figure S8). For the fine mode, proton-promoted processing accounts for approximately 39% of the soluble iron deposition which is about six times higher than oxalate-promoted processing.

A discernible decline in dust soluble iron deposition to the NWP was found over the study period, with an annual reduction rate of approximately 2.4% per year. Dust soluble iron deposition decreased by 31% from 2001 to 2017 spring. Among them, both initially emitted and atmospheric processing promoted soluble iron deposition showed a decreasing trend. From 2001 to 2017 spring, the deposition of emitted dust soluble iron, proton-promoted dust soluble iron, and oxalate-promoted dust soluble iron over the NWP decreased by 37%, 5%, and 24%, respectively. Also, both coarse and fine mode soluble iron deposition exhibited a decreasing trend (approximately 30%).

However, the amount of soluble iron deposition produced from atmospheric processing showed a much lower decrease (18%) in 2017 compared to 2001 spring (Figure S9). The coarse-mode proton-promoted soluble iron deposition even increased by 7% as shown in Fig. S9d. This finding indicates that atmospheric processing may offset the decline of dust emission to some extent regarding soluble dust iron deposition over NWP. Fig. 5(b) also suggested that despite of the decrease in absolute amounts of dust total and soluble iron deposition ("Total Fe"), the proportion of soluble iron in total dust iron was found to increase over the study period ("soluble/Total Fe"). The proportion increased from 1.5% in 2001to 1.7% in 2017. This increase has contributed an additional 0.4 Gg/season of soluble iron to the NWP in 2017 (8% of the average values). We further probed into the ratios of soluble iron produced by proton-promoted and oxalate-promoted processes in total dust iron, and found they both increased prominently (Figure S10). The increased coarse mode proton-promoted ratio and oxalate-promoted ratio induced the increase of iron solubility (Figure S10). Our results suggested that atmospheric processing contributed to the increase in dust iron solubility and mitigated the overall decline trend in dust soluble iron deposition, indicating key factors such as aerosol acidity and in-cloud oxalate.

### 3.3 Contributions from different driving factors to change of dust iron

To reveal the underlying drivers behind the long-term trend of soluble dust iron deposition, in this section we investigated the changes in dust emission, aerosol-pH, and in-cloud oxalate concentrations while the further contributions of anthropogenic SO$_2$ and NO$_x$ emissions were quantified through sensitivity simulations.

### 3.3.1 Dust emissions and surface wind

We first examined the decreasing trend of dust aerosol concentration over EA and found a strong correlation between surface wind speed and dust emission. The Gobi (38-50N, 100-120E) and Taklamakan (35-43N, 80-100E) deserts are primary sources of dust emissions in EA. Our simulation revealed that the average dust emission flux was 2.3 and 6.5 Gg/day in spring over Gobi and Taklamakan, respectively, and surface wind speed was 4.9 and 4.0 m/s, respectively (Fig. 6c and Fig. 6d). Notably, the correlation coefficients between dust emission flux and surface wind speeds at the Gobi and Taklamakan were 0.7 and 0.6 respectively, underscoring surface wind speed as a pivotal factor in controlling dust emissions. An evident declining trend in surface wind speeds was observed at both dust sources, leading to reduced dust emissions and burdens. This finding is consistent with previous studies (Guan et al., 2017; Wu et al., 2022; Xu et al., 2020) that also reported the dominant role of surface wind speed in the dust reduction trend in East Asia. Specifically, Guan et al. (2017) and Xu et al. (2020) focused on dust storm events and emphasized the dominant role of maximum surface wind speed based on observed datasets. Our study provided a direct relationship between dust emissions and surface wind speed. Compared to the modelling study by Wu et al. (2022) which mainly focused on the Gobi Desert, we illustrated the dust surface wind's role over the two dust sources including Gobi and Taklamakan Desert. Moreover, the impact of varying wind directions on dust emission and transport was evident. Chen et al. (2017) reported that dust from Gobi was predominantly driven by westerly winds, exerting a greater influence on downwind eastern inland areas and the North Pacific. This relationship was corroborated by our results, which presented more pronounced similarities in the variations of dust deposition to the NWP with dust emissions from the Gobi source compared to the Taklamakan source.

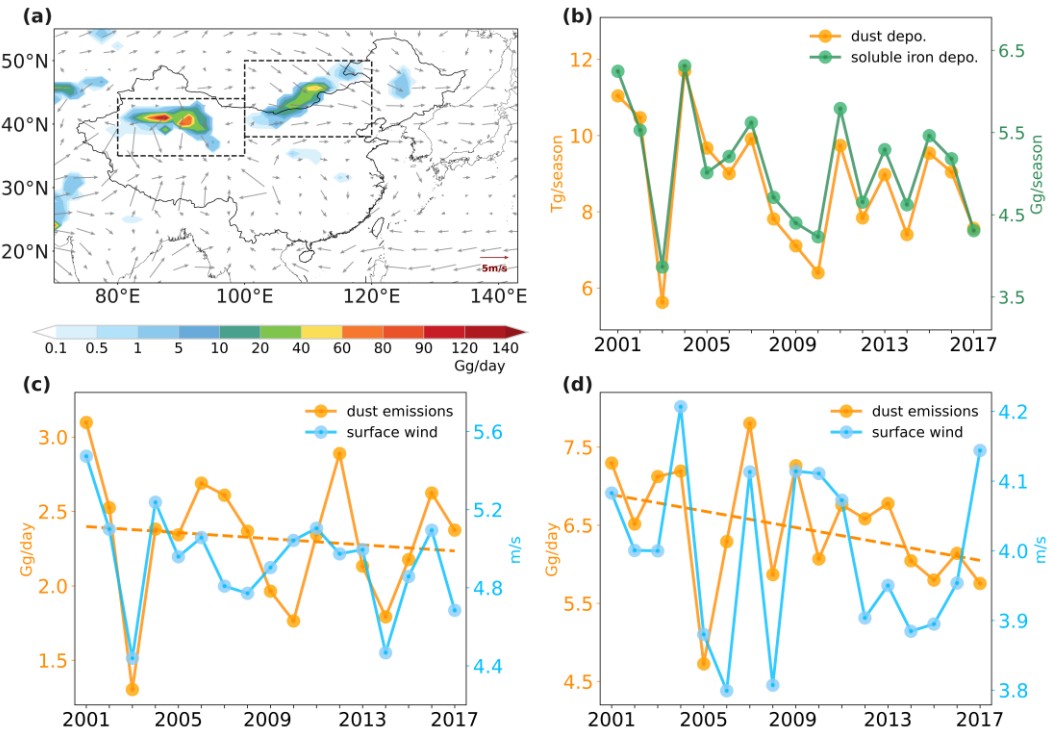

**Figure 6: (a) Spatial distributions of dust emission flux and surface wind over Eastern Asia during 2001-2017 springs. (b) Temporal variations of dust (yellow) and soluble iron (green) deposition to Northwest Pacific (30-50N, 140E-160W). (c, d) Temporal variations of dust emission flux (yellow) and surface wind (blue) over the Gobi dust source (38-50N, 100-120E) and Taklamakan dust source (35-43N, 80-100E) on average of 2001-2017 springs.**

Influenced by the decreased surface wind and dust emissions over two dust sources, dust deposition to the NWP showed a

prominent decreasing trend of approximately 2% per year (yellow line in Fig. 6b). This decrease in dust deposition has induced

the decline in dust soluble iron deposition (green line in Fig. 6b). The temporal variations in dust soluble iron deposition were

governed by the variations of dust deposition of which the correlation coefficient was high to 0.96. In conclusion, our findings

highlight that the decrease in soluble iron depositions to the NWP can be attributed to the diminished dust emissions which

was induced by declined surface wind speeds over dust sources.

**3.3.2 Aerosol pH and anthropogenic emissions**

We found an increasing trend in the solubility of dust iron despite of the declines in both total and soluble dust iron depositions

in the NWP, indicating the effects of atmospheric processing may also change over the study period. So, this section probes

into the trend of atmospheric processing and the related variables.

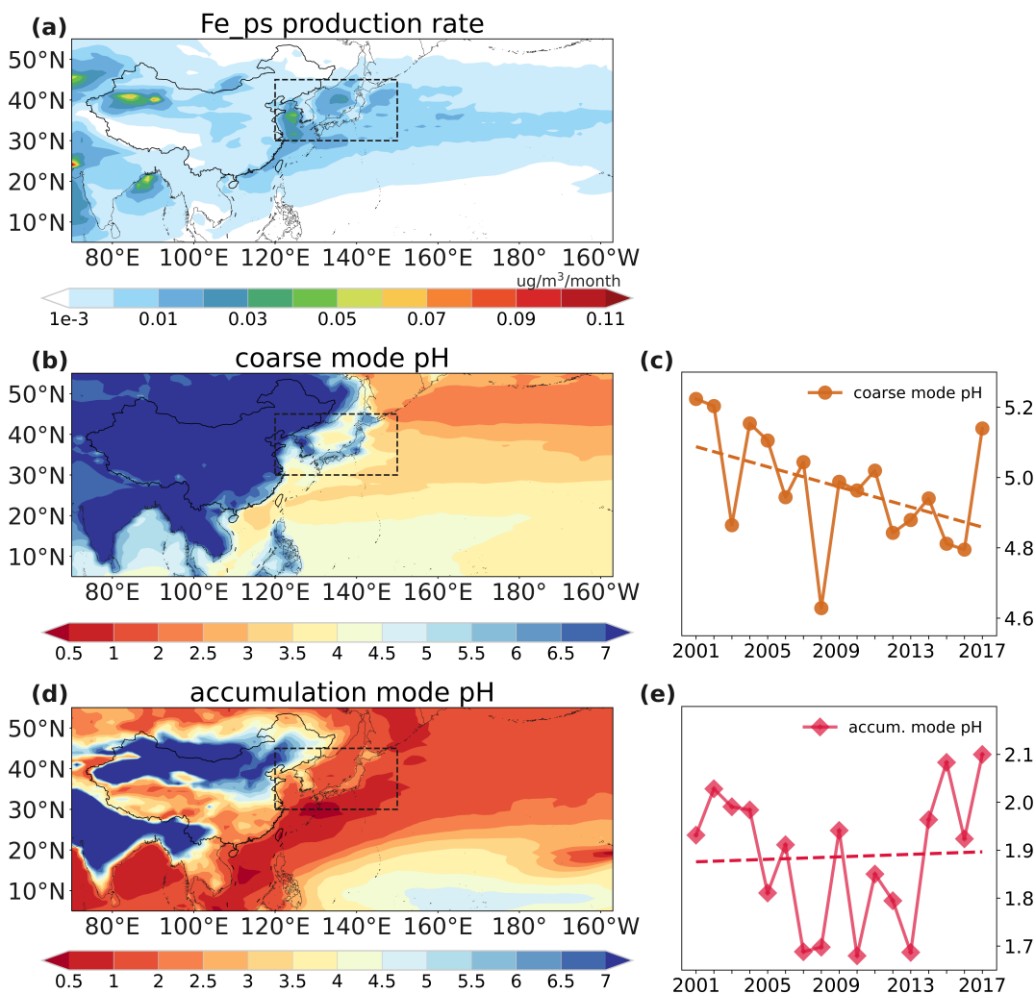

Figure 7: Spatial distributions of surface dust soluble iron production rate from proton-promoted processing (a), surface coarse mode aerosol pH (b), and surface accumulation mode aerosol pH (d) averaged of 2001-2017 springs. Temporal variations of surface coarse mode pH (c) and accumulation mode pH (e) over high production rate area (30-45N, 120-150E) averaged of 2001-2017 springs.

We first explored the proton-promoted processing where aerosol pH is a key factor influencing soluble iron production. The spatial distribution of surface dust soluble iron production rate from proton-promoted processing was shown in Fig. 7(a). We find through model simulation that the soluble iron production rate from atmospheric processing primarily peaked at the surface layer of the model (Figure S11). The most significant production was found over downwind areas of North China Plain over the Bohai Sea and the Huanghai Sea (shown in dashed rectangles in Fig. 7), which are referred to as high production area hereafter. The production of proton-promoted soluble iron is mainly determined by aerosol pH and dust total iron concentrations (Eq.5). Hence high production rate area was characterized by both relatively high dust concentrations and elevated aerosol acidity (Fig. 7b and Fig. 7d). We analyzed aerosol pH trend over the high production area of the proton-promoted which would have a major influence on soluble iron deposition over the NWP (Fig. 7c and Fig. 7e). We found in

the simulation that pH values showed different trend in coarse and fine mode aerosols. For coarse mode, the averaged pH values in this area demonstrated a noticeable decreasing trend, indicating an enhanced intensity of proton-promoted soluble iron over the study period. For accumulation mode aerosol, however, the pH value showed a minor increasing trend (Fig. 7e).

To understand the different trends in coarse and fine aerosol pH, we further investigated the key factors (such as acidic gases and $CaCO_3$) affecting aerosol pH through the MOSAIC mechanism. In the dust source area, $CaCO_3$ dominated both coarse and accumulation modes of dust. Compared to coarse mode, the content of dust would be relatively lower in accumulation mode while the content of sulfate, nitrate, and hydrochloride aerosols would be higher. As for coarse mode pH, it was mostly weakly alkaline according to Eq.1 (pH > 7 when T < 25°C) during spring over land areas in EA due to sufficient $CaCO_3$ content

(Fig. 7b) during spring over land areas in EA (Fig. S12a). The simulated quasi-neutral pH of continental coarse mode aerosols has been confirmed by thermodynamic models using aerosol samples, as reported by Fang et al. (2017) and Ding et al. (2019). During the transportation, the solid $CaCO_3$ would be consumed by acidic gases ($H_2SO_4$, $HNO_3$, and $HCl$) which would produce $CaSO_4$, $Ca(NO_3)_2$ and $CaCl_2$ (Figure S12). And coarse mode aerosol pH became more acidic in the coastal and ocean areas. The acidic pH of oceanic coarse mode aerosols which ranged from 2-5 agreed with the estimated results from mineral dust

particles (Meskhidze et al., 2003). We find through model simulation that in coarse mode, $H_2SO_4$ was almost depleted to produce insoluble $CaSO_4$, a condition termed sulfate-poor (Eq.2, $X_T > 2$). Therefore, in areas of high proton-promoted production, coarse mode aerosol pH was determined by semi-volatile acidic gases ($HNO_3$ and $HCl$) following Eq.4. As for accumulation mode aerosol pH, it exhibited higher acidity due to relatively lower $CaCO_3$ content (Fig. 7d). We find through model simulation that in accumulation mode, sufficient sulfate content resulted in sulfate-rich condition (Eq.2, $X_T < 2$).

Therefore, the emissions of $SO_2$ have a more pronounced effect on accumulation mode aerosols pH following Eq.3, while it has a very limited impact on coarse model aerosols pH.

Increased coarse aerosol acidity could be attributed to enhanced $NO_x$ and $HCl$ concentrations (Figure S13) according to the abovementioned functions within the MOSAIC mechanism. Anthropogenic $NO_x$ emission in China increased significantly over 2001-2014, and previous studies also reported the subsequent enhancement of $HNO_3$ especially over North China Plain

with intensive $NO_x$ emission (Luo et al., 2020b). And the increasing trend in HCl concentration was induced by the enhanced $HNO_3$ gas which would produce HCl gases through heterogeneous reactions with $NaCl/CaCl_2$. The increased trend has also been observed along the coast of East Asia (Gromov et al., 2016). Furthermore, decreased dust emissions would also enhance the aerosol acidity due to the less consumption of acidic gases by $CaCO_3$ as reported by Karydis et al. (2021). As for accumulation mode, the reduction in anthropogenic $SO_2$ emissions was expected to decrease aerosol acidity whereas the

reduction in dust emissions could conceivably lead to an increase in aerosol acidity. Within this mechanism, the accumulation mode pH showed a very minor decreasing trend. Our findings are in agreement with observational studies highlighting the roles of $SO_4^{2-}$ and non-volatile cations ($Ca^{2+}$) in determining fine particle acidity which only presented minor changes in EA recently (Ding et al., 2019; Zhou et al., 2022).

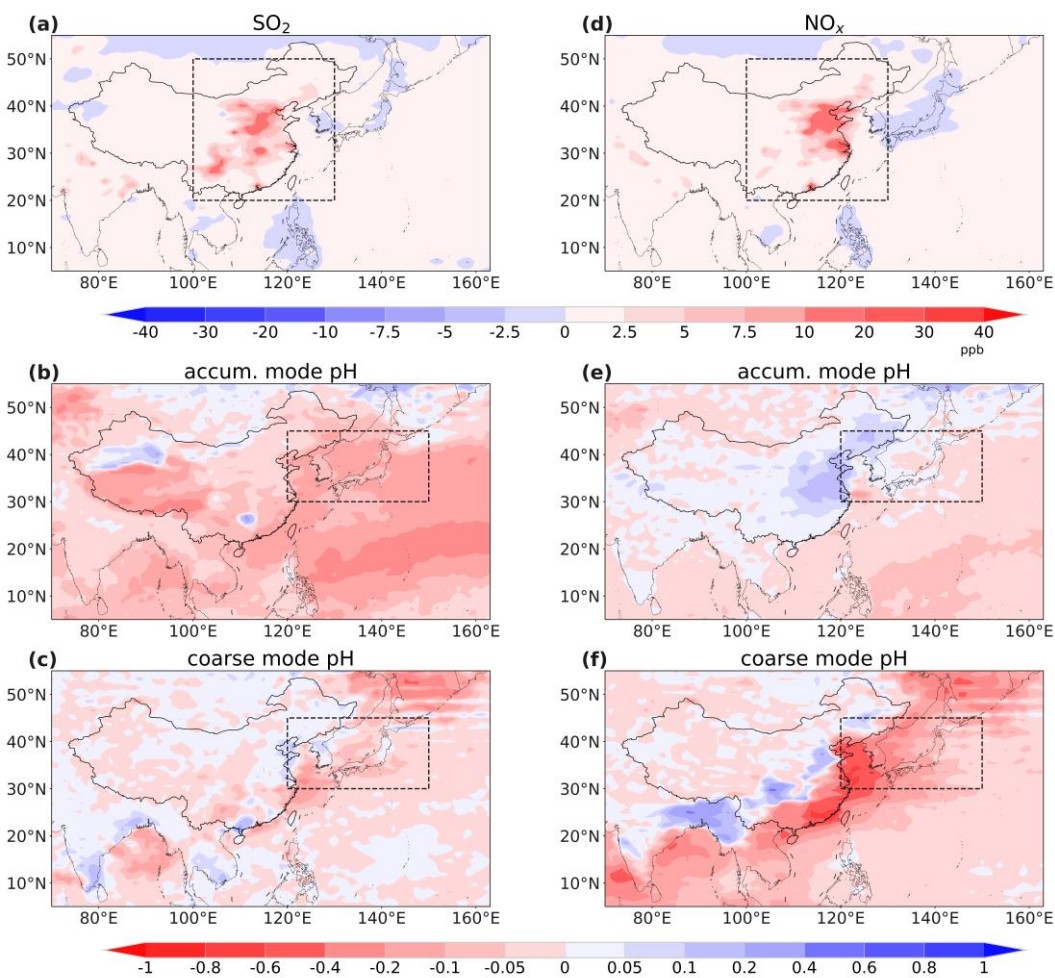

**Figure 8: Spatial distributions of surface concentration changes of SO₂ (a) and NOₓ (d) in sensitivity experiments and the induced coarse mode aerosol pH (b and e) and accumulation mode aerosol pH (c and f) variations respectively.**

Anthropogenic SO$_2$ emission was usually believed to play an important role in determining aerosol pH and driving the proton-promoted solubility as reported in previous studies (Li et al., 2017b; Fan et al., 2006), but our simulation results indicated a dominant role by NO$_x$ which driving the long-term trend of dust iron solubility change. To further consolidate it, two sensitivity experiments were conducted (as illustrated in Table 1) to demonstrate anthropogenic NO$_x$ emission indeed has a more significant influence on proton-promoted soluble iron deposition than SO$_2$. We used a higher SO$_2$ emission for simulation scenario "SO$_2$ change" to estimate the impact of SO$_2$ emission change on soluble iron by comparing it with baseline simulation, and similarly for scenario "NO$_x$ change". Compared to the baseline simulation of 2001, concentrations of both SO$_2$ and NO$_x$ have all increased by about 2 ppb averaged in the east and central part of China (the square area dash outlined in Fig. 8a and Fig. 8d). However, the increase in SO$_2$ didn't induce obvious changes in coarse mode aerosol pH while the increase in NO$_x$ notably decreased coarse mode aerosol pH, especially over coastal areas. In the proton-promoted high production area (dash

outlined in Fig. 8c and Fig. 8f), coarse mode aerosol pH changed -0.03 and -0.2 while accumulation mode aerosol pH changed -0.2 and +0.02 induced by enhanced $SO_2$ and $NO_x$ respectively. The decreased fine mode aerosol acidity over the mainland of EA could be attributed to the increased aerosol water content induced by enhanced $NO_x$ (Figure S14). As a result, the soluble iron deposition to the NWP increased by 0.01 and 0.03 Gg/season induced by $SO_2$ and $NO_x$ respectively. Compared with $SO_2$ emission which mainly affects accumulation mode, $NO_x$ has a greater impact on proton-promoted soluble iron by affecting coarse mode which dominated (as shown in section 3.2). Our findings underscored the significant impact of anthropogenic $NO_x$ over $SO_2$ on dust soluble iron deposition historically. This result is consistent with the study by Ito and Xu (2014) which showed that future (in 2100) reductions in NOx which were based on the RCP4.5 would lead to a decrease in soluble iron deposition over North Pacific (40–60°N, 140–230°E). However, our model also indicated that changes in $SO_2$ emissions would also contribute to dust soluble iron deposition changes (approximately one-third of the impact of $NO_x$) their model showed that $SO_2$ had a negligible effect.

In summary, the increase in iron solubility could be attributed to the enhanced acidity of coarse mode aerosols driving by anthropogenic $NO_x$ emission change. The interplay of increased $NO_x$/HCl concentrations and reduced dust emissions has led to an upward trend in coarse mode aerosol acidity. Conversely, the reduction in $SO_2$ and dust emissions have counterbalanced their respective effects on fine mode aerosol acidity, which showed insignificant trends. Our findings suggest that mitigating acidic gas emissions, particularly $NO_x$, could decelerate dust iron dissolution by influencing aerosol acidity. Furthermore, the reduction of dust emissions would increase aerosol acidity and the induced soluble iron production. Such proton-promoted processing would mitigate the decreasing trend of soluble deposition induced by decreased initial emissions.

### 3.3.3 Oxalate and cloud environment

We further analyze the key factors (oxalate concentration and cloud fraction) of oxalate-promoted processing. Oxalate concentration has an important effect on the oxalate-promoted soluble iron production according to Eq.6. In addition, the oxalate-promoted processing takes place only when iron aerosols are in the cloud-borne phase in our model. It has been observed the enhanced iron dissolution when complexation with aerosols occurs in moderately acidic conditions such as cloud environment (Cornell and Schindler, 1987; Paris et al., 2011; Xu and Gao, 2008). This setting is consistent with the previous studies (Scanza et al., 2018; Hamilton et al, 2019) which based on the experiment study of Paris et al. (2011). Specifically, the cloud environment represents the cloud fraction is higher than 1E-5 and cloud liquid water content (LWC) is higher than 1E-8 (L/L(cloud)) in the model. The possibility of oxalate-promoted processing would increase if the cloud fraction and LWC increase. Hence, we focus our analysis on two factors including oxalate concentration and cloud fraction which is a proxy of cloud environment in this section. The spatial distributions of oxalate concentration and cloud fraction are shown in Fig9. (b) and Fig9. (d) respectively. The spatial pattern of the simulated surface oxalate concentration pattern aligned with the other simulations and in-cloud observations over EA (Myriokefalitakis et al. 2022; Zhao et al., 2019).

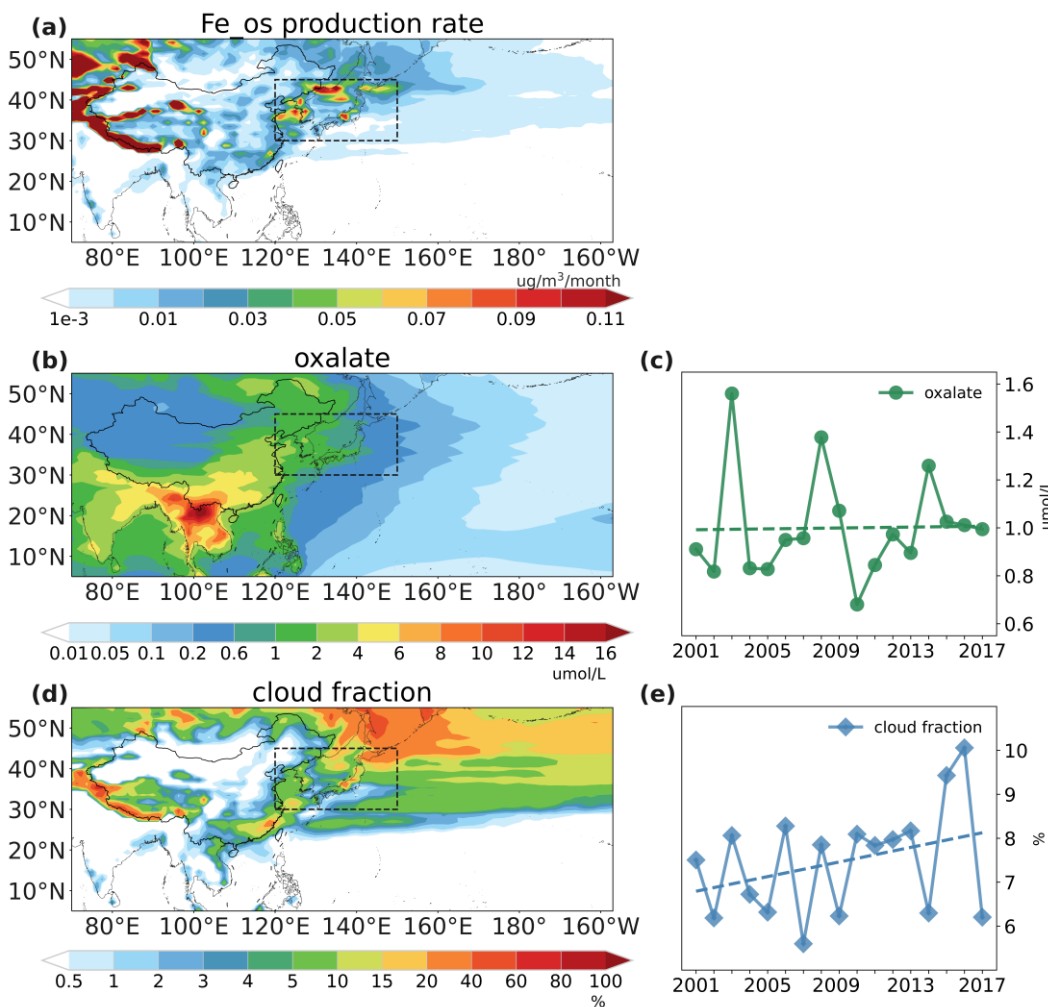

**Figure 9: Spatial distributions of surface dust soluble iron production rate from oxalate-promoted processing (a), surface oxalate concentrations (b), and surface cloud fractions (d) averaged of 2001-2017 springs. Temporal variations of surface oxalate concentrations (c) and cloud fractions (e) over high production rate area (30-45N, 120-150E) averaged of 2001-2017 springs.**

As shown in Fig. 9(a), the high production rate of oxalate-promoted processing also peaks in the same area (30-45N, 120-150E) as the proton-promoted processing. Both surface oxalate concentrations (Fig. 9b) and cloud fractions (Fig. 9d) were relatively higher in this area which contributed to the high production of oxalate-promoted processing. The averaged oxalate concentration in the high production rate area didn't show an obvious increasing trend (Fig. 9c). This finding was consistent with the long-term observations which presented a tiny increasing trend of oxalate over the western North Pacific (Boreddy et al., 2017). This indicated that oxalate concentration didn't drive the change of oxalate-promoted dust iron. We found in the simulation that the surface cloud fraction in the high production rate area enhanced prominently as shown in Fig. 9(e). And the increase of cloud fraction over EA has also been reported by previous studies utilizing ground-based and satellite

observations (Huang et al., 2020; Yang et al., 2023). Clouds are influenced by numerous intricate cloud-controlling factors such as relative humidity, surface temperature and estimated inversion strength (Kawai and Shige, 2020; Klein et al., 2017). The increase of surface cloud fraction in the high production rate area may be related to the increase of relative humidity (Figure S15) which plays a significant role in the study of long-term clouds variation (Cao et al., 2021; Yang et al., 2023). Similarly, the temporal LWC was also increased as shown in Figure S16. The increased cloud fractions and LWC would

provide more possibility for oxalate-promoted processing as it only occurs in the cloud-borne phase, thereby enhancing oxalate-promoted soluble iron production.

## 3.4 Chlorophyll-a response

Our study also integrated simulated soluble iron deposition with satellite chlorophyll-a observations to investigate the chlorophyll-a response to soluble iron deposition. We found a positive correlation between elevated soluble iron deposition

and increased chlorophyll-a levels in NWP over the study period. We calculated the relative change in chlorophyll-a levels based on the 4-day period preceding and following each soluble iron deposition event. This analysis method was proposed by Westberry et al. (2023) and was demonstrated to successfully characterize the relationship between iron deposition and chlorophyll-a response. This 4-day interval balances the episodic nature of dust events and the typical response time of phytoplankton, which is generally less than one week. We identified the highest soluble iron deposition events (HDEs) on an

interannual scale for each grid point each spring. The remaining daily data from the 16-year period were categorized as low deposition events (LDEs). We found in the simulation that soluble iron deposition during HDEs is nearly 8 times that of LDEs in the NWP (Fig. 10a and Fig. 10b).

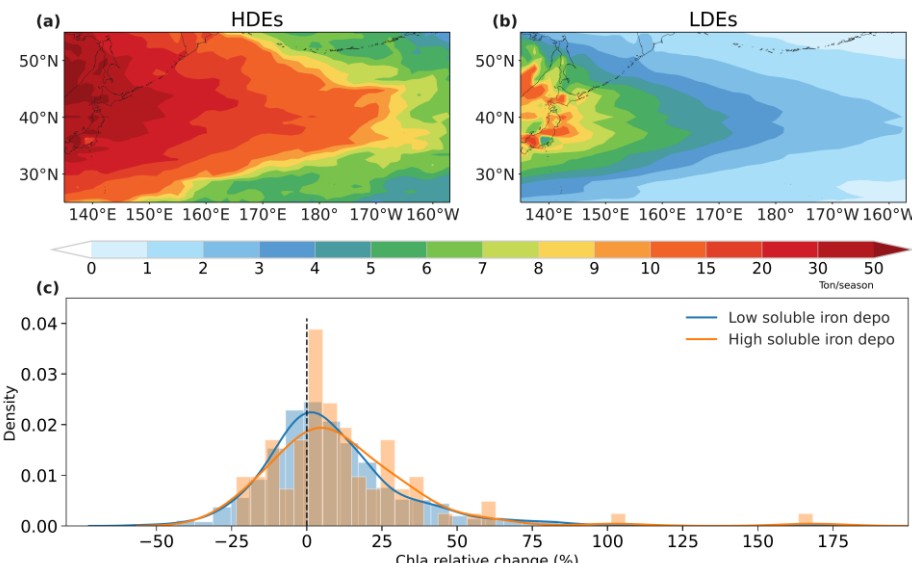

**Figure 10: Spatial distributions of selected high (a) and low (b) soluble iron deposition averaged of 2001-2017 springs. (c) probability**
**distribution function of relative changes in chlorophyll-a averaged over NWP after high (orange) and low(blue) deposition events.**

The probability distribution function of chlorophyll-a relative changes after HDEs and LDEs events over the NWP region is used to further illustrate the influence of dust iron deposition, as shown in Fig. 10(c). Given the typical spring bloom of phytoplankton, the calculated relative responses were predominantly positive. Post-HDE responses showed a more pronounced shift towards positive values compared to LDEs. HDEs resulted in an increased probability of chlorophyll-a growth (72% compared to 61% for LDEs). Furthermore, the average relative change in chlorophyll-a following HDEs was greater than that following LDEs (12% vs. 9%). This probability distribution pattern implied that stronger soluble dust iron inputs can enhance phytoplankton growth over NWP. Previous studies also reported an increase in phytoplankton growth after intensive dust deposition in the NWP (Zhang et al., 2023b; Luo et al., 2020; Yoon et al., 2017). However, it is important to notice that phytoplankton dynamics are also influenced by various other factors, including sea surface temperature, mixed layer depth, and upwelling transport. To isolate the impact of dust soluble iron on phytoplankton, more targeted experiments and simulations are required such as applying a fully coupled earth system model.

## 4 Conclusions and summaries

This study employs an advanced modeling approach to provide a long-term analysis of the spatiotemporal dynamics and driving factors influencing dust soluble iron deposition in the Northwest Pacific in the spring of 2001-2017. We performed simulations of dust total and soluble iron emissions, transport, and deposition with the CAM6-Chem model. The developed model incorporated a mineralogy map of dust iron emissions and accounted for two primary atmospheric processing—proton-promoted and oxalate-promoted mechanisms—that induce iron dissolution. Validated against multiple observational datasets, our model successfully captured the characteristics of total and soluble iron over the North Pacific.

We found through the simulation that the average deposition of dust soluble iron to the Northwest Pacific was 4.9 Gg/season. Atmospheric processing played a significant role (~40%) in dust soluble iron deposition of which the oxalate-promoted processing emerged as a dominant contributor (25%). A decline in surface wind strength over dust source regions in East Asia led to a 2.4% annual decrease in dust soluble iron deposition. However, the proportion of dust soluble iron within the total deposition has been rising despite the general decrease in both dust total and soluble iron deposition. This increase has contributed an additional 0.4 Gg/season of soluble iron to the NWP in 2017 (8% of the average values). On the one hand, the increase in iron solubility can be attributed to enhanced $NO_x$/HCl emissions and reduced dust emissions, which increased coarse mode aerosol acidity and iron dissolution. On the other hand, the increased surface cloud fractions accelerated the oxalate-promoted processing and induced more dust soluble iron production. Atmospheric processing has thus been identified as a crucial factor in promoting soluble iron production and deposition in the Northwest Pacific. Furthermore, our findings indicate that anthropogenic $NO_x$ emissions exert a more significant influence on the dissolution of dust soluble iron through proton-promoted processing rather than $SO_2$. As for the marine ecology of dust soluble iron deposition, our study establishes a link between high dust soluble iron depositions and increased chlorophyll-a levels in the NWP. It highlights the ecological

significance of soluble iron inputs in promoting phytoplankton growth and the potential ecology effect induced by declined dust soluble iron deposition in the Northwest Pacific.

It is crucial to acknowledge that this study focuses on spring dust sources of iron but pyrogenic iron sources, such as those from anthropogenic activities and biomass burning in other seasons also make a substantial contribution to the ocean's soluble iron inventory due to their high solubility (Ito et al., 2021; Ito et al., 2019; Rathod et al., 2020). The increased anthropogenic soluble iron deposition trend during our study period could opposite the decreased dust soluble iron deposition to some extent (Hamilton et al., 2020). Future studies should include these additional iron sources to provide a more comprehensive assessment of soluble iron and its ecological impacts throughout the year. Additionally, further investigation into other atmospheric processing that facilitates iron dissolution, such as irradiation effects (Faust and Hoigné, 1990; Spokes and Jickells, 1995) and the influence of various organic matter ligands (Chen and Grassian, 2013; Sakata et al., 2022), is necessary. The lack of an explicit chemical simulation for oxalate in our study highlights the need for continued development in understanding iron dissolution mechanisms, aiming to improve the model's ability to simulate soluble iron in dust accurately. Given the limitations of the Community Earth System Model (CESM2) in simulating dust dynamics (Wu et al., 2020), urgent advancements are needed in modeling dust emissions, transport, and deposition processes to enhance the accuracy of dust iron simulations (Li et al., 2022; Leung et al., 2023).

In summary, our research delineates a quantitative decrease in dust bioavailable iron deposition in the Northwest Pacific which may have broader ecological consequences. We emphasize the importance of atmospheric processing including the proton-promoted and oxalate-promoted mechanisms in the dissolution of dust iron. Both initial emissions of land and atmospheric processing play an important role in dust bioavailable iron deposition and subsequent marine ecology effect. Amidst the evolving global environmental challenges, a holistic understanding of the complex interactions in earth system and the impact of human activities is essential.

**Code and data availability**

The code and data of our developed model will be made available on request.

**Author contribution**

**Hanzheng Zhu**: Conceptualization, Formal analysis, Investigation, Methodology, Visualization, Writing – original draft, Writing – review & editing. **Yaman Liu**: Methodology. **Man Yue**: Methodology. **Shihui Feng**: Formal analysis, Writing – review & editing. **PingQing Fu**: Formal analysis, Writing – review & editing. **Kan Huang**: Formal analysis, Writing – review & editing. **Xinyi Dong**: Conceptualization, Formal analysis, Investigation, Project administration, Supervision, Writing – original draft, Writing – review & editing. **Minghuai Wang**: Project administration, Supervision, Writing – review & editing.

**Competing interests**

At least one of the (co-)authors is a member of the editorial board of Atmospheric Chemistry and Physics.

**Acknowledgments**

This work acknowledges the financial support from the National Natural Science Foundation of China (Grants 41925023,

42075102, U2342223, and 42361144711), and the Fundamental Research Funds for the Central Universities - CEMAC "GeoX" Interdisciplinary Program (2024ZD05) by the Frontiers Science Center for Critical Earth Material Cycling, Nanjing University, as well as the Collaborative Innovation Center of Climate Change, Jiangsu Province. We also appreciate the High-Performance Computing Center of Nanjing University for providing the computational resources essential for this research. Special thanks are extended to all the scientists, software engineers, and administrators involved in the development of the CESM2 model.

We thank for the observations data from GEOTRACES. The GEOTRACES 2021 Intermediate Data Product version 2 (IDP2021v2) represents an international collaboration and is endorsed by the Scientific Committee on Oceanic Research (SCOR). The many researchers and funding agencies responsible for the collection of data and quality control are thanked for their contributions to the IDP2021v2. Additionally, we are grateful to the anonymous reviewers for their valuable feedbacks and suggestions aimed at enhancing the quality of the manuscript.

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
