# Peer review of "Trends and Drivers of Soluble Iron Deposition from East Asian Dust to the Northwest Pacific: A Springtime Analysis (2001-2017)"

_EGUsphere, 2024_

## Author Comment (AC2)

**Reply for the referee comment#2**

**General comments**

Model predictions of chemical composition in dust aerosols and its effect on iron solubility and ocean biogeochemistry are highly uncertain. The authors implemented a mineralogical map and atmospheric processing schemes of dust to a global atmospheric chemistry model. They evaluated the model results against observations of total and soluble iron concentrations. They conducted sensitivity experiments to assess the impact of recent shifts in dust emissions and chemical compositions on soluble iron deposition in the Northwest Pacific. Their results indicate a decreasing trend in dust soluble iron deposition from East Asia to the Northwest Pacific by 2.4% per year, primarily due to reduced dust emissions, which are mainly driven by declining surface winds over dust source regions. They show an increasing trend in dust iron solubility from 1.5% in 2001 to 1.7% in 2017. This increased iron solubility is associated with the acidification of coarse mode aerosols due to the increase in anthropogenic $NO_x$ emissions and in-cloud oxalate-ligand-promoted dissolution. The modeling exercises in this paper may help us to advance modeling dust iron. However, the differences of their model development from previous studies are unclear in its current form. It is more appropriate to cite parent papers rather than the following papers to clarify the model development. I have some comments and questions to improve this paper.

**General response**: We would like to express our gratitude for your detailed and insightful feedback. Below, we provide responses to each comment and indicate how the manuscript will be revised accordingly.

**Specific comment#1**: l.89: How did you calculate $Na^+$ on dust?

**Response**: Thank you for your insightful comment. We didn't calculate $Na^+$ on dust. The $Na^+$ in Eq.3a are mainly sourced from sea-salt aerosol and the concentration is calculated for each aerosol modes but not the specified aerosol type. In our study, the CAM-chem model employed the MOSAIC module to calculate aerosol pH for four aerosol modes: Aitken, Accumulation, Coarse, and Primary Carbon. Different types of aerosols, such as sulfate, nitrate, dust, and sea-salt, are externally mixed between the modes and internally mixed within each mode (Liu et al., 2016). The ion concentration is calculated for each mode and this includes both dust and sea-salt aerosols. $Na^+$ in the model primarily originates from sea-salt aerosols, which are represented by electrolytes such as $Na_2SO_4$, $NaNO_3$, and $NaCl$.

Reference

Liu, X., Ma, P. L., Wang, H., Tilmes, S., Singh, B., Easter, R. C., Ghan, S. J., and Rasch, P. J.: Description and evaluation of a new four-mode version of the Modal Aerosol Module (MAM4) within version 5.3 of the Community Atmosphere Model, Geosci. Model Dev., 9, 505-522, 10.5194/gmd-9-505-2016, 2016.

**Specific comment#2**: l.98 and Fig. 7: What are the criteria using either (4a) or (4b)? Please elucidate how alkaline cations as in (3a) are considered for pH calculation in the sulfate-poor conditions to understand the acidic conditions in coarse particles over the oceans (see below comments on Fig. 7).

$$m_{H^+} = 2m_{SO_4^{2-}} + m_{HSO_4^-} + m_{NO_3^-} + m_{Cl^-} - (2m_{Ca^{2+}} + m_{NH_4^+} + m_{Na^+}) \tag{3a}$$

$$m_{H^+} = \frac{K^{gl}_{HNO_3} C_{l,HNO_3}}{\kappa_{HNO_3} m_{NO_3^-} (\gamma_{HNO_3})^2}, or \tag{4a}$$

$$m_{H^+} = \frac{K^{gl}_{HCl} C_{l,HCl}}{\kappa_{HCl} m_{Cl^-} (\gamma_{HCl})^2} \tag{4b}$$

**Response**: The criteria for using equations (4a) and (4b) depend on the equilibrium concentrations of nitrate and chloride in both the gas and aerosol phases. Specifically, equation (4a) is applied when $HNO_3$ gas and $NO_3$ aerosol are both present and have concentrations greater than zero while Equation (4b) is used when HCl gas and chloride aerosol concentrations are greater than zero after computing equilibrium surface concentrations.

In sulfate-poor conditions, it is important to account for the gas-aerosol exchange of semi-volatile gases such as $HNO_3$, HCl, and $NH_3$. Using the internal equilibrium H+ concentration like Eq.3 would lead to oscillations in the condensation and evaporation of these gases, as it does not provide steady-state results (Zaveri et al., 2008). Therefore, MOSAIC employs dynamic H+ concentrations which is determined by equilibrium constants, mass transfer coefficients, and the gas- and particle-phase concentrations of all involved semi-volatile species like Eq.4. Alkaline cations as described in Eq.3a have no effect on the pH calculation in sulfate-poor conditions. We have specified the pH mechanism in line 102.

*Line 102:* "Under sulfate-poor conditions ($XT > 2$), it is important to account for the gas-aerosol exchange of semi-volatile gases such as $HNO_3$, HCl, and $NH_3$. Using the internal equilibrium H+ concentration would lead to oscillations in the condensation and evaporation of these gases, as it does not provide steady-state results (Zaveri et al., 2008). Therefore, MOSAIC employs dynamic H+ concentrations which is determined by gas-particle exchange of semi-volatile acidic gases ($HNO_3$ and HCl) predominantly controls aerosol acidic pH as follows (Eq.4):

…

calculated by MOSAIC (Zaveri et al., 2008; Zaveri et al., 2005a; Zaveri et al., 2005b). Specifically, equation (4a) is applied when $HNO_3$ gas and $NO_3$ aerosol are both present and have concentrations greater than zero while Equation (4b) is used when HCl gas and chloride aerosol concentrations are greater than zero after computing equilibrium surface concentrations."

Reference

Zaveri, R. A., Easter, R. C., Fast, J. D., and Peters, L. K.: Model for Simulating Aerosol Interactions and Chemistry (MOSAIC), Journal of Geophysical Research: Atmospheres, 113, https://doi.org/10.1029/2007JD008782, 2008.

**Specific comment#3**: l.112: This mineralogy map has been implemented by previous studies. Please cite the parent papers rather than your references and elucidate the differences from previous studies.

**Response**: Thank you for your insightful comment. We have specified the reference paper and added a discussion in the line 124.

*Line 124:* "In this study, a detailed mineralogy map database (Nickovic et al., 2012) was implemented into the CAM6-chem model to configure the mineral composition of dust emissions. This map is based on the work of Claquin et al. (1999) and has been widely used in previous modeling studies (Johnson and Meskhidze, 2013; Ito and Xu, 2014; Myriokefalitakis et al., 2015). The map segments soil into silt/clay fractions and includes five main iron minerals: hematite, smectite, illite, kaolinite, and feldspar. Notably, compared to the mineralogy map from Journet et al. (2014), this map has been shown to perform well, particularly in identifying phyllosilicates with high soluble iron content (Gonçalves Ageitos et al., 2023)."

**Specific comment#4**: Please specify iron content and initial iron solubility in minerals.

**Response**: Thanks for your insightful comment. We have added Table S2 to provide the iron content and initial iron solubility for each of the five minerals. The statement is in line 135.

*Table S2. Iron content and initial iron solubility for each of the five minerals*

|  | Fe_rs | Fe_ms | Fe_ss | Fe content | Fe solubility |
|---|---|---|---|---|---|
| **Hematite** | 0% | 0% | 57.5% | 57.5% | 0% |
| **Smectite** | 0.55% | 10.45% | 0% | 11.0% | 5% |
| **Illite** | 0.11% | 3.89% | 0% | 4.0% | 2.8% |
| **Kaolinite** | 0.01% | 0% | 0.23% | 0.24% | 4.2% |
| **Feldspar** | 0.01% | 0% | 0.33% | 0.34% | 2.9% |

*Line 135:* "On top of the mineralogy, …with proportions aligned with Hamilton et al. (2019) and Scanza et al. (2018). The detailed iron content and initial iron solubility for each of the five minerals have been shown in Table S2."

**Specific comment#5**: l.119: This scheme has been implemented by previous studies. Please cite the parent papers rather than your references and elucidate the differences from previous studies.

**Response**: Thanks for your insightful comment. We have now included the parent papers and expanded the discussion in line 135.

*Line 135:* "On top of the mineralogy, …with proportions aligned with Hamilton et al. (2019) and Scanza et al. (2018). The detailed iron content and initial iron solubility for each of the five minerals have been shown in Table S2. These values are sourced from measurements (Journet et al., 2008; Shi et al., 2011a; Shi et al., 2011b) and are consistent with previous modeling studies (Ito and Xu 2014; Scanza et al., 2018; Hamilton et al., 2019)"

**Specific comment#6**: l.122, Figure S1: Please show the comparison of iron content with observations to prove the underestimations of dust iron in main dust source regions by default settings and discuss the major reasons.

**Response**: Thanks for your insightful comment. We have added the global distribution of initial iron content in dust coarse mode aerosol as Figure S1a and discussed it in the article in line 138.

*Line 138:* "According to the utilization of the mineralogy map, our model achieved to simulate the global spatial patterns of total and initial soluble iron emissions. Compared to the default setting of 3.5%, the total iron content in dust aerosol is higher in the main dust sources including North Africa, Middle East and central Asia, and East Asia (Fig. S1a). This is consistent with the observations (Lafon et al., 2004, 2006; Shi et al., 2011b) and the research by Ito and Xu (2014), which reported that the observed iron content in North Africa and East Asia averaged 3.7%. Therefore, the use of the mineralogy map increases the iron content in dust from these regions (Fig. S1b) which suggest the default settings likely underestimate dust iron in these main dust source regions."

[Figure]

*Figure S1. (a) Spatial distribution of iron content in coarse mode dust aerosol. (b) Compared to the default setting (3.5% iron in dust), changes in dust total iron surface concentrations from the developed model averaged 2001-2017 springs.*

**Specific comment#7**: l.124: The two atmospheric processing has been implemented by previous studies. Please cite the parent papers rather than your references and elucidate the differences from previous studies.

**Response**: Thanks for your insightful comment. We have added the parent papers and discussed the differences from previous studies that included atmospheric processing in line 145 and line 165.

*Line 145:* "For proton-promoted dissolution, we employ the first-order dissolution rate formula of Lesaga et al. (1994) as follows (Eq.5) for interstitial iron aerosols:

$$RFe_{i,proton} = K_i(T) \times a(H^+)^{m_i} \times f(\nabla G_r) \times A_i \times MW_i \qquad (5a)$$

$$\frac{d}{dt}\left[Fe_{sol,\ proton}\right] = RFe_{i,proton} \times [Fe_{insol}] \qquad (5b)$$

…, A is the specific surface area and MW is the molecular weight. The parameters for the first-order dissolution rate formula (Eq. 5) are based on previous studies (Meskhidze et al., 2003; Ito and Feng, 2010; Ito and Xu, 2014) and are aligned with Scanza et al. (2018) and Hamilton et al. (2019). For $K_i(T)$ in Fe_ms type, here we use the dissolution rate of mineral illite as an additional simplification following Scanza et al., (2018). In contrast, some studies (Ito and Feng, 2010; Ito, 2012; Ito and Xu,

2014) employ separate dissolution rates for different minerals. For $K_i(T)$ in Fe_ss type, we use a fast dissolution rate with three stages, following Ito and Xu (2014)."

*Line 165:* "For oxalate-promoted dissolution, we employ the first-order dissolution rate formula for cloud-borne iron aerosols as follows (Eq.6):

$$RFe_{i,oxalate} = a_i \times [C_2O_4^{2-}] + b_i \qquad (6a)$$

$$\frac{d}{dt}\left[Fe_{sol,\ oxalate}\right] = RFe_{i,oxalate} \times [Fe_{insol}] \qquad (6b)$$

…, coefficients $a_i$ and $b_i$ are determined by Paris et al., (2011) and aligned with Scanza et al., (2018) and Hamilton et al., (2019). This linear relationship between oxalate-promoted dissolution rate and oxalate concentration in solution is based on cloud water studies by Paris et al. (2011) and has been employed in previous modeling studies (Johnson and Meskhidze, 2013; Myriokefalitakis et al., 2015). Furthermore, Ito and Shi (2016) developed a new oxalate-promoted scheme, which has been applied in recent research (Myriokefalitakis et al., 2022)."

**Specific comment#8**: l.135, Figure S2: Did you use the annually averaged pH values? Please rephrase observed pH, because there is no direct measurement of pH in aerosol. Since this study focuses on coarse mode aerosols, why don't you show the similar figure for the coarse mode to support the model performance?

**Response**: Thanks for your insightful comment. The annually averaged pH values were used in this study. We have replaced "observed" with "observationally estimated" to clarify that there are no direct measurements of pH in aerosols. The lack of sufficient observations makes it challenging to provide a detailed comparison for the model performance with respect to coarse mode aerosols. We have made revisions as follows.

*Line 158:* "The simulated aerosol pH in accumulation and coarse mode have been shown in Figure S2. The fine particles are relatively more acidic while coarse-mode particles are significantly less acidic influenced by sea salt and dust components. Through the annually averaged comparison of accumulation mode aerosols' pH with observations collected by Pye et al. (2020), our model successfully captured the global characteristics of fine aerosol pH. The correlation coefficient and normalized mean bias (NMB) are 0.4 and 27% respectively. The discrepancy could be attributed by the seasonal variations and the dynamics of precursor gas emissions, environmental factors such as relative humidity."

[Figure]

*Figure S2. (a) Spatial distribution of aerosol pH in accumulation mode in 2013 and observationally estimated ground-level fine-aerosol pH (dots) from Pye et al. (2020). (b) The linear relationship between aerosol pH simulation and observationally estimated ground-level fine-aerosol pH. (c) Spatial distribution of aerosol pH in coarse mode in 2013.*

**Specific comment#9**: l.147 and section 3.3.3: Please show the comparisons with observed oxalate levels in East Asia cloud water.

**Response**: We have gathered global oxalate observations in rain/cloud water to provide a more detailed comparison. The references have been listed in supplementary. Figures S4 in the supplementary material show the spatial distribution of observation locations and the evaluation of the simulation. We have expanded the oxalate comparison in line 182 as follows:

*Line 181:* "For the oxalate evaluation, we have collected global oxalate observations in rain/cloud water to evaluate our model results. The locations and months are consistent between observations and the model. Comparisons with observed oxalate levels indicate that our model accurately captures the quantitative characteristics of oxalate especially in East Asia (Figure S4)."

[Figure]

*Figure S4. (a) Sample locations of the observed oxalate in rain/cloud water (Sempéré and Kawamura, 1996; Willey et al., 2000; Brooks Avery et al., 2001; Kawamura et al., 2001; Hegg et al., 2002; Kieber et al., 2002; Löflund et al., 2002; Peña et al., 2002; Kim et al., 2003; Sigha-Nkamdjou et al., 2003; Crahan et al., 2004; Hu et al., 2005; Brooks Avery et al., 2006; Xu et al., 2009; Huang et al., 2010; Huo et al., 2010; Sumari et al., 2010; Gioda et al., 2011; Wang et al., 2011; Zhang et al., 2011; Khuntong, 2012; Zhu et al., 2016; Du et al., 2017; Zhao et al., 2019;*

*Zhang et al., 2021; González et al., 2022; Lee et al., 2022; Xie et al., 2022; Sun et al., 2024). (b) The comparison between estimated oxalate concentration in cloud water and observations. The read circles are East Asia sites and the cyan circles are not East Asia sites.*

**Specific comment#10**: l. 215, Figure 3: Please show the comparison of iron solubility.

**Response**: We have added the comparison of iron solubility as Figure S7. Our model only captures the 0-10% iron solubility range. This is likely due to the lack of pyrogenic iron which has been suggested to contribute to higher iron solubility (Ito et al., 2019). We have added a discussion in line 268.

*Line 267*: "What's more, the comparison about iron solubility between simulation and observations has shown in Figure S7. Our model only captures the 0-10% iron solubility. This is likely due to the lack of pyrogenic iron which has been suggested to contribute to higher iron solubility (Ito et al., 2019)."

[Figure]

*Figure S7. The comparison about iron solubility between simulation and observations.*

Reference

Ito, A., Myriokefalitakis, S., Kanakidou, M., Mahowald, N. M., Scanza, R. A., Hamilton, D. S., Baker, A. R., Jickells, T., Sarin, M., Bikkina, S., Gao, Y., Shelley, R. U., Buck, C. S., Landing, W. M., Bowie, A. R., Perron, M. M. G., Guieu, C., Meskhidze, N., Johnson, M. S., Feng, Y., Kok, J. F., Nenes, A., and Duce, R. A.: Pyrogenic iron: The missing link to high iron solubility in aerosols, Science Advances, 5, eaau7671, doi:10.1126/sciadv.aau7671, 2019.

**Specific comment#11**: l.255: There is no reference of observational research to support this modelling results. Please correct the sentence or add the reference if any.

**Response**: Thanks for your insightful comment. In the study by Shi et al. (2022), observational data were used to develop a deep learning neural network (DLNN) model. Through this model, oxalate was identified as the most significant variable influencing iron solubility. We have specified the reference to this observational study in line 322.

*Line 322*: "And this result is consistent with previous modelling (Johnson and Meskhidze, 2013; Scanza et al., 2018) and observation research (Shi et al., 2022)."

Reference

Shi, J., Guan, Y., Gao, H., Yao, X., Wang, R., and Zhang, D.: Aerosol Iron Solubility Specification

in the Global Marine Atmosphere with Machine Learning, Environmental Science & Technology, 56, 16453-16461, 10.1021/acs.est.2c05266, 2022.

**Specific comment#12**: l.266, l.272 and Figure S5: It is not clear whether coarse-mode proton-promoted soluble iron deposition increased by 7% from Fig. 5. Please show the statistics for the increased trends. Please show the trend of total soluble iron deposition clearly to elucidate atmospheric processing contributed to the increase in dust iron solubility.

**Response**: Thanks for your insightful comment. We have added the annual trend of proton-promoted and oxalate-promoted soluble iron in fine and coarse mode as Figure S9. We expanded the discussion in line 340. We also added the linear trend in Figure S10 to show atmospheric processing contribution. We have expanded the discussion in line 340 and line 348.

*Line 340:* "However, the amount of soluble iron deposition produced from atmospheric processing showed a much lower decreasing rate by 18% (Figure S9). The coarse-mode proton-promoted soluble iron deposition even increased by 7% as shown in Fig. S9d."

*Line 348:* "We further probed into the ratios of soluble iron produced by proton-promoted and oxalate-ligand-promoted processes in total dust iron (Figure S10). The increased coarse mode proton-promoted ratio and oxalate-promoted ratio induced the increase of iron solubility (Figure S10). Our results suggested that atmospheric processing contributed to the increase in dust iron solubility..."

[Figure]

*Figure S9. Temporal variations of the ratio of dust soluble iron deposition from proton-promoted (a, d), oxalate-ligand-promoted (b, e) and emissions (c, f) in coarse and fine mode (atiken + accumulation) to the Northwest Pacific averaged of 2001-2017 springs.*

[Figure]

*Figure S10. Temporal variations of the ratio of dust soluble iron deposition from proton-promoted (a, d), oxalate-ligand-promoted (b, e) and emissions (c, f) in coarse and fine mode (atiken + accumulation) to dust total iron deposition to the Northwest Pacific averaged of 2001-2017 springs.*

**Specific comment#13**: l.281: The decreasing trend of dust aerosol concentration over EA and a strong correlation between surface wind speed and dust emission has been shown in previous studies, as you cited. Please discuss the similarities and differences between this and previous studies quantitatively.

**Response**: Thanks for your insightful comment. We have expanded the discussion in line 364.

*Line 364:* "This finding is consistent with previous studies (Guan et al., 2017; Wu et al., 2022; Xu et al., 2020) that also reported the dominant role of surface wind speed in the dust reduction trend in East Asia. Specifically, Guan et al. (2017) and Xu et al. (2020) focused on dust storm events and emphasized the dominant role of maximum surface wind speed based on observed datasets. Our study provided a direct relationship between dust emissions and surface wind speed. Compared to the modelling study by Wu et al. (2022) which mainly focused on the Gobi Desert, we illustrated the dust surface wind's role over the two dust sources including Gobi and Taklamakan Desert."

**Specific comment#14**: l.320 and Fig. 7: Since the first-order mass transfer coefficient is used to simulate size-dependent pH, $HNO_3$ absorption in the coarse mode should occur slowly during the long-range transport. Thus, it is unclear why pH is suddenly dropped once the aerosols are transported over the oceans. Please specify the mechanisms of quickly elevated aerosol acidity over the oceans. It is also unclear how calcite in coarse particles is consumed. Please show the spatial distribution of excess Ca (in the form of $CaCO_3$ in MOSAIC), $CaSO_4$, $Ca(NO_3)_2$ and $CaCl_2$ and specify the mechanisms of acidification of coarse particles.

**Response:** Thank you for your insightful comment. The first-order mass transfer coefficient is lower in the coarse mode due to the higher radius. And it would result in a slower absorption of $HNO_3$ during long-range transport. However, the rapid drop in pH once the coarse aerosols are transported over the oceans can be explained by several factors. Firstly, the higher relative humidity in the marine atmosphere would increase the water content of coarse mode aerosols as shown in the figure blow. This would enhance the mean wet radius ($\overline{R}_{p.m}$) of coarse mode aerosol which would enhance the first-order mass transfer coefficient ($k_{i,m}$) of $HNO_3$ and HCl.

$$k_{i,m} = 4\pi \overline{R}_{p,m} D_{g,i} N_m f_{i,m}$$

where $k_{i,m}$ is the mass transfer rates for species i and aerosol bin m, $\overline{R}_{p,m}$ is the mean wet radius of aerosols in bin m, $D_{g,i}$ is gas diffusivity of species i, $N_m$ is the number concentration of aerosols in bin m, and $f_{i,m}$ is the transition regime correction factor. Hence, the enhanced transfer of acidic gases induced the increased acidity of coarse mode aerosol. Secondly, pH in MOSAIC is calculated for each aerosol mode and the coarse mode is significantly influenced by sea-salt aerosols. The increased presence of sea-salt aerosols in the coarse mode over the oceans would also promote the acidification of aerosols. Thus, the combination of higher humidity and sea-salt aerosol concentration leads to a rapid drop in pH when coarse mode aerosols reach oceanic regions.

[Figure]

*Figure. Spatial distribution of aerosol water content in coarse mode averaged of 2001 spring.*

And the calcite in the particle is consumed by the following irreversible heterogeneous reactions (R1, R4, and R5) shown in Table S1. And the spatial distribution of solid-phase $CaCO_3$, solid-phase $CaSO_4$, liquid-phase $Ca(NO_3)_2$, and liquid-phase $CaCl_2$ averaged of 2001 spring has been shown in Figure S12. As dust was transported, the coarse mode solid $CaCO_3$ was consumed by $HNO_3(g)$, $H_2SO_4(g)$, and $HCl(g)$ which produced $CaSO_4$, $Ca(NO_3)_2$, and $CaCl_2$ respectively. Specially, the $CaCO_3$ was mainly consumed by $HNO_3(g)$ which produced $Ca(NO_3)2$ due to the much higher $HNO_3$ gas than $H_2SO_4$ gas. And the maximum of $CaSO_4$ in the coastal region ([35-40N, 120-140E]) could be attributed to the R2-3 in Table S1.

We have specified the pH mechanism in line 87.
***Line 87:*** "Firstly, the MOSAIC mechanism would determine whether the aerosol contains solid $CaCO_3$ which can adsorb acidic gases ($H_2SO_4$, $HNO_3$, and HCl). The irreversible heterogeneous reactions which would consume solid $CaCO_3$ have been listed in Table S1."

We have also further explained the acidification of coarse particles over East Asia in line 328.
***Line 407:*** "As for coarse mode pH, it was mostly weakly alkaline according to Eq.1 (pH > 7 when T < 25°C) during spring over land areas in EA (Fig. 7b) due to the sufficient solid $CaCO_3$ content during spring over land areas in EA (Fig. S12a). The simulated quasi-neutral pH of continental coarse mode aerosols has been confirmed by thermodynamic models using aerosol samples, as reported by Fang et al. (2017) and Ding et al. (2019). During the transportation, the solid $CaCO_3$ would be consumed by acidic gases ($H_2SO_4$, $HNO_3$, and HCl) which would produce $CaSO_4$, $Ca(NO_3)_2$ and $CaCl_2$ (Figure S12). And the coarse mode aerosol pH coarse mode aerosol pH became more acidic in the coastal and ocean areas."

*Table S1. List of irreversible heterogeneous reactions about Ca. (Zaveri et al., 2008)*

| | **Irreversible heterogeneous reactions** |
|---|---|
| **R1** | $CaCO_3(s) + H_2SO_4(g) \rightarrow CaSO_4(s) + H_2O(g) + CO_2(g)$ |
| **R2** | $CaCl2(s,l) + H_2SO_4(g) \rightarrow CaSO_4(s) + 2HNO_3(g)$ |
| **R3** | $Ca(NO_3)_2(s,l) + + H_2SO_4(g) \rightarrow CaSO_4(s) + 2HCl(g)$ |
| **R4** | $CaCO_3(s) + 2HNO_3(g) \rightarrow Ca(NO_3)_2(s) + H_2O(g) + CO_2(g)$ |
| **R5** | $CaCl2(s,l) + 2HNO_3(g) \rightarrow Ca(NO_3)_2(g) + 2HCl(g)$ |
| **R6** | $CaCO_3(s) + 2HCl(g) \rightarrow CaCl_2(s) + H_2O(g) + CO_2(g)$ |

[Figure]

*Figure S12. Spatial distributions of coarse mode solid-phase CaCO₃(a), solid-phase CaSO₄(b), liquid-phase Ca(NO₃)₂, and liquid-phase CaCl₂ averaged of 2001 spring.*

**Specific comment#15**: l.334: Please compare with the dust aerosols, instead of sea spray aerosols cited in this sentence.

**Response**: Thanks for your insightful comment. We have changed to the reference about the dust aerosol acidity in line 413.

*Line 413:* "And coarse mode aerosol pH became more acidic in the coastal and ocean areas. The acidic pH of oceanic coarse mode aerosols which ranged from 2-5 agreed with the estimated results from mineral dust particles (Meskhidze et al., 2003)."

Reference

Meskhidze, N., Chameides, W. L., Nenes, A., and Chen, G.: Iron mobilization in mineral dust: Can anthropogenic $SO_2$ emissions affect ocean productivity?, Geophysical Research Letters, 30, https://doi.org/10.1029/2003GL018035, 2003.

**Specific comment#16**: l.341, l.445: Please explain why HCl emissions and concentrations were increased?

**Response**: The gas HCl in the model is not from primary emissions but secondary source. In MOSAIC, the HCl gas are from irreversible heterogeneous reactions between more acidic gases (such as $HNO_3$) and salt of chloride (NaCl and $CaCl_2$). It is the increased $NO_x$ emissions induced

higher $HNO_3$ gas and then more HCl are product. The increased $HNO_3$ gas has been shown in Figure S13. We have expanded the discussion in line 424 and corrected the earlier term of HCl emissions in line 458.

*Line 424:* "And the increasing trend in HCl concentration was induced by the enhanced $HNO_3$ gas which would produce HCl gases through heterogeneous reactions with $NaCl/CaCl_2$. The increased trend has also been observed along the coast of East Asia (Gromov et al., 2016)."

*Line 458:* "On the one hand, the increase in iron solubility can be attributed to enhanced $NO_x/HCl$ concentrations and reduced dust emissions"

[Figure]

*Figure S13. Temporal variations of surface concentrations of $SO_2$ (a), $NO_x$ (a), HCl (c), and $HNO_3$(d) over the high production rate of proton-promoted soluble iron area (30-45N, 120-150E) averaged of 2001-2017 springs.*

**Specific comment#17**: l.370: The dominant role of $NO_x$ in the long-term trend of dust iron solubility has been shown in previous studies, as you cited. Please discuss the similarities and differences between this and previous studies quantitatively.

**Response**: Thanks for your insightful comment. In the study of Ito and Xu (2014), the impact of future changes in anthropogenic emissions of $NO_x$ and $SO_2$ on dust soluble iron deposition to the subarctic North Pacific (40–60°N, 140–230°E) was explored. The reduction of $NO_x$ emissions in 2100 based on the RCP4.5 compared to 2000 would lead to 15% decrease in dust soluble iron deposition promoted by acid mobilization. The changes of $SO_2$ emissions had negligible effect on dust soluble iron deposition. In this study, the impact of historical changes of $NO_x$ and $SO_2$ emissions in 2007 spring compared to 2001 spring on dust soluble iron deposition to the Northwest Pacific (30-50°N, 140–200°E) was explored. The increased $NO_x$ and $SO_2$ emissions induced 4% and 1% increased proton-promoted dust soluble iron deposition. We all have clarified the dominated role of $NO_x$ other than $SO_2$ on dust soluble iron deposition. In addition, our model also indicated that changes in $SO_2$ emissions would also contribute to dust soluble iron deposition changes (approximately one-third of the impact of $NO_x$) but their model showed that $SO_2$ had a negligible effect. We have discussed it in line 451.

*Line 451:* "Our findings underscored the significant impact of anthropogenic $NO_x$ over $SO_2$ on dust

soluble iron deposition historically. This result is consistent with the study by Ito and Xu (2014) which showed that future (in 2100) reductions in $NO_x$ which were based on the RCP4.5 would lead to a decrease in soluble iron deposition over North Pacific (40–60° N, 140–230° E). However, our model also indicated that changes in $SO_2$ emissions would also contribute to dust soluble iron deposition changes (approximately one-third of the impact of $NO_x$) their model showed that $SO_2$ had a negligible effect."

**Specific comment#18**: l.386: The oxalate-promoted dissolution of iron in acidic solution is much faster than cloud water conditions according to laboratory experiments. However, the oxalate-promoted dissolution of iron in aerosol is not considered in this study. Please specify the most rapid conditions.

**Response**: Thanks for your insightful comment. We deleted the 'most' description and specified the occurring environment in line 466.

*Line 466:* "The oxalate-promoted processing takes place only when iron aerosols are in the cloud-borne phase in our model. It has been observed the enhanced iron dissolution when complexation with aerosols occurs in aqueous solutions under moderately acidic conditions such as clouds environment (Cornell and Schindler, 1987; Paris et al., 2011; Xu and Gao, 2008). This setting is consistent with the previous studies (Scanza et al., 2018; Hamilton et al, 2019) and based on the experiment study of Paris et al. (2011)."

**Technical comments:**

(1) l.85: Please indicate the unit of T.

**Response**: Thanks for your insightful comment. We have indicated it line 90.
*Line 90*: "The process was influenced by temperature (T) of which the unit is K as follows (Eq.1):"

(2) l.90: Please indicate the unit of m.

**Response**: Thanks for your insightful comment. We have indicated it line 95.
*Line 95*: "where m is the concentration of ions in aerosol water and the unit is mol/kg"

(3) l.95: Please indicate the unit of K. Please explain the activity coefficient used in (3b).

**Response**: Thanks for your insightful comment. We have indicated it in line 101.
*Line 101*: "where $K_{HSO4}$ is the equilibrium constant of the bisulfate ion dissociation and the unit is $mol^2\,kg^{-2}\,atm^{-1}$ and $\gamma$ is the activity coefficient of electrolyte calculated by MOSAIC"

(4) l.100: Please indicate the unit of C.

**Response**: Thanks for your insightful comment. We have indicated it line 110.
*Line 110*: "$C_l$ is the equilibrium concentrations of acidic matter for the liquid phase and the unit is $mol/m^{-3}$ (air)"

(5) l.173, Figure S4, and Figure 3: Please specify GEOTRACES, cite the parent papers, and follow the Fair Use Agreement (Fair_Data_Use_Statement-for-IDP2021v2.pdf).

**Response**: Thanks for your insightful comment. We have specified it.

*Line 258*: "GEOTRACES Intermediate Data Product Group (2021)"
*Line 579*: "We thank for the observations data from GEOTRACES. The GEOTRACES 2021 Intermediate Data Product version 2 (IDP2021v2) represents an international collaboration and is endorsed by the Scientific Committee on Oceanic Research (SCOR). The many researchers and funding agencies responsible for the collection of data and quality control are thanked for their contributions to the IDP2021v2"
*Line 643*: "GEOTRACES Intermediate Data Product Group (2021). The GEOTRACES Intermediate Data Product 2021 version 2 (IDP2021v2). NERC EDS British Oceanographic Data Centre NOC. doi: 10.5285/ff46f034-f47c-05f9-e053-6c86abc0dc7e"

*Line 46 in supplementary*: "GEOTRACES Intermediate Data Product Group (2021)"
*Line 120 in supplementary*: "GEOTRACES Intermediate Data Product Group (2021). The GEOTRACES Intermediate Data Product 2021 version 2 (IDP2021v2). NERC EDS British Oceanographic Data Centre NOC. doi: 10.5285/ff46f034-f47c-05f9-e053-6c86abc0dc7e"

(6) l.432: Please correct to Fig. 10 (c).

**Response**: Thanks for your insightful comment. We have corrected it line 511.
*Line 511*: "The probability distribution function of chlorophyll-a relative changes after HDEs and LDEs events over the NWP region is used to further illustrate the influence of dust iron deposition, as shown in Fig. 10(c)."

---

## Author Comment (AC3)

**Reply for the referee comment#1**

**Summary:** This study examines the deposition of soluble iron from dust aerosols using the Community Atmosphere Model version 6 (CAM6-chem). CAM6-chem has been developed here to include desert dust mineralogy and to incorporate proton- and oxalate-promoted dissolution schemes for the iron-containing dust aerosols. The main focus of this work is on the factors influencing the deposition of soluble iron from dust in the Northwest Pacific during the spring seasons from 2001 to 2017, with evaluation against observational datasets from the North Pacific. The authors report a decrease in the deposition of soluble iron from East Asia to the Northwest Pacific during the studied period, which is attributed to reduced dust emissions. However, they also observe an increase in dust iron solubility, primarily linked to the atmospheric processing of coarse dust aerosols. Sensitivity simulations indicate that rising anthropogenic NOx emissions, rather than a reduction in SO2, are the primary factor influencing dust aerosol acidity in the model, leading overall to an increase in iron solubility despite the decrease in iron from dust. The manuscript is well-written; however, some issues concerning the methodology and the presentation of results should be addressed before final publication. This will help readers better understand the assumptions considered in this work along with the uncertainties surrounding the main conclusions derived from model simulations.

Summary Response: We sincerely appreciate the detailed feedback, which has significantly contributed to improving our manuscript. Below, we provide responses to each comment and describe the corresponding revisions.

**General comment#1:** The authors used the global model CAM6-chem to simulate the soluble iron deposition over the Northwest Pacific. Given that a number of global modelling studies provide global budget calculations of the atmospheric iron cycle (i.e., burdens, wet and dry deposition rates, iron solubilisation rates, etc.), both for total and soluble iron or per mode (fine and coarse) iron-containing aerosols, the authors should provide their global estimations along with those for the study area. I also propose to present the budget calculations in a separate table and present other modelling estimates for comparison.

Response for comment#1: Thank you for your insightful comment. We have added global and Northwest Pacific (NWP) atmospheric iron cycle budget in Table 2. This table includes burdens, wet and dry deposition rates, and iron solubilization rates for both fine and coarse dust total/soluble iron aerosols calculated from 2017 all year modelling. Additionally, we have compared our model results with the Mechanism of Intermediate complexity for Modelling Iron (MIMI) from Hamilton et al. (2019) of which the period is 2007-2011, EC-Earth model from Myriokefalitakis et al. (2022) of which the period is 2000-2014, and the ensemble modeling study of Myriokefalitakis et al. (2018) of which the period across 2007-2014. The comparison results have been shown in Table 3. We discuss the atmospheric iron cycle in line 279.

*Line 279:*
"3.2 Spatial and temporal characteristics of dust iron deposition
3.2.1 Atmospheric dust iron budget

…

[revised manuscript text omitted]

**General comment#2:** In Sect. 2, the calculation of oxalate concentrations in the model used for the ligand-promoted dissolution is not clearly explained. The authors employed the formula from Hamilton et al. (2019) to estimate atmospheric oxalate levels based on the modeled secondary organic carbon concentrations. However, Hamilton et al. (2019) established a maximum aqueous concentration threshold of 15 μmol L$^{-1}$, derived from the estimations of Scanza et al. (2018). What threshold is applied here? Do the authors calculate with their model version similar secondary organic carbon concentrations as reported by Scanza et al. (2018) and Hamilton et al. (2019)? If it differs, what threshold was used? Additionally, how might this assumption affect the simulated oxalate concentrations?

**Response**: Thank you for your insightful comment. Below, we address the key points raised:

*(1) Threshold Consistency*

We employed the same maximum aqueous oxalate concentration threshold of 15 μmol L$^{-1}$, consistent with Hamilton et al. (2019) and Scanza et al. (2018).

*(2) Comparison of Secondary Organic Aerosols (SOA) Models*

Hamilton et al. (2019) employed the CAM versions 5 and 6 (CESM-CAM5–6; Neale et al., 2010). In our study, the version CAM6-Chem coupled with MOSAIC was used.

In CAM5 and CMA6, SOA is simulated through the pre-calculated, lumped SOA gas-phase species undergoing reversible condensation and evaporation into aerosols. This gas-phase SOA precursor (SOAG) is derived from fixed mass yields for five categories of volatile organic compounds (VOCs) with yields increased by 50% after tuning for aerosol indirect effects (Neale et al., 2010; Liu et al., 2012).

In CAM6-Chem, SOA formation follows a Volatility Basis Set (VBS) approach with explicit VOCs and chemistry (Emmons et al., 2020; Tilmes et al., 2019). It incorporates wall-corrected SOA yields, photolytic removal of SOA, and more efficient removal by dry and wet deposition. In addition, the CAM6-Chem model applied in our study also simulates the heterogeneous uptake of isoprene epoxydiols (IEPOX) onto sulfate aerosols and production of IEPOX-SOA (Jo et al., 2019; 2021).

The simulated SOA burden was 1.15 Tg/yr in CAM5 (2001-2006; Liu et al., 2012), 1.07 Tg/yr in CAM6 (1995-2010; Tilmes et al., 2019) and 1.02 Tg/yr in CAM6-Chem (2013; Jo et al., 2023). In terms of SOA mechanism used by our study, the SOA burden was 1.42 Tg/yr during 2001-2017 which is little higher but comparable with the 1.15 Tg/yr in CAM5 as used by Hamilton et al. (2019).

Furthermore, the spatial distribution has been shown in Figure S3 and it is comparable with the results (Fig. 3a) from CAM5 (Liu et al., 2012). Specifically, the maximum secondary organic carbon concentrations calculated in our model during 2001-2017 is close to the number (1.41 vs 1.41 mol/mol) used in Hamilton et al. (2019). Based on this consistency, we chose not to alter the parameters in the formula from Hamilton et al. (2019). As a result, the scaled oxalate concentrations align well with the observational data especially in East Asia as shown in Figure S4.

*(3) Impact on Oxalate Estimation and future work*

Differences in SOA concentrations in our study could lead to overestimation of oxalate concentrations and the subsequent oxalate-promoted soluble iron contribution. However, Hamilton et al. (2019) did not analyze the relative contributions of proton-promoted and oxalate-promoted processes. This makes it challenging to determine whether oxalate-promoted contributions are consistently over- or underestimated in our study. We acknowledge this limitation and emphasize the need for direct oxalate concentration calculations in future work to improve simulation accuracy.

We have made a further discussion in line 80 and line 179.

**Line 80:**
"In CAM6-Chem, SOA formation follows a Volatility Basis Set (VBS) approach with explicit VOCs and chemistry (Emmons et al., 2020; Tilmes et al., 2019). It incorporates wall-corrected SOA yields, photolytic removal of SOA, and more efficient removal by dry and wet deposition. What's more, the heterogeneous uptake of isoprene epoxydiols (IEPOX) onto sulfate aerosols and their subsequent production are explicitly simulated through coupling with MOSAIC (Jo et al., 2019; 2021)."

**Line 179:**
"The threshold of oxalate concentration is 15 μmol $L^{-1}$ keeping consistent with Hamilton et al. (2019) and Scanza et al. (2018). Because the SOA burden simulated in our model version (Figure S3) is comparable with the previous version (Fig. 5a; Liu et al., 2012). The maximum SOA concentration was similar to the study of Hamilton et al. (2019). For the oxalate evaluation, … our model accurately captures the quantitative characteristics of oxalate especially in East Asia (Figure S4)."

[Figure]

*Figure S3. Spatial distribution of Secondary Organic Aerosols (SOA) burden from 2001 to 2017 with simulation data from Liu et al. (2023).*


**General comment#3:** The authors note that the model accurately captures oxalate observations. However, in Sect. 3.3.3, only the simulated surface oxalate concentration patterns over EA are presented, with no evaluation of the modeled OXL concentrations against observations. As far as I understand, the authors only compare spatial patterns from other modeling studies. I suggest that the authors present an evaluation of their model using observations (both globally and with a special focus on the EA region), as done in the other studies referenced in the manuscript.

**Response**: Thanks for the comment. We collected global oxalate observations in rain/cloud water to provide a more detailed comparison. The references have been listed in supplementary. Figures S4 in the supplementary material show the spatial distribution of observation locations and the evaluation of the simulation. We have expanded the oxalate comparison in line 182 as follows:

*Line 181:* "For the oxalate evaluation, we have collected global oxalate observations in rain/cloud water to evaluate our model results. The locations and months are consistent between observations and the model. Comparisons with observed oxalate levels indicate that our model accurately captures the quantitative characteristics of oxalate especially in East Asia (Figure S4)."

[Figure]

*Figure S4. (a) Sample locations of the observed oxalate in rain/cloud water (Sempéré and Kawamura, 1996; Willey et al., 2000; Brooks Avery et al., 2001; Kawamura et al., 2001; Hegg et al., 2002; Kieber et al., 2002; Löflund et al., 2002; Peña et al., 2002; Kim et al., 2003; Sigha-Nkamdjou et al., 2003; Crahan et al., 2004; Hu et al., 2005; Brooks Avery et al., 2006; Xu et al., 2009; Huang et al., 2010; Huo et al., 2010; Sumari et al., 2010; Gioda et al., 2011; Wang et al., 2011; Zhang et al., 2011; Khuntong, 2012; Zhu et al., 2016; Du et al., 2017; Zhao et al., 2019; Zhang et al., 2021; González et al., 2022; Lee et al., 2022; Xie et al., 2022; Sun et al., 2024). (b) The comparison between estimated oxalate concentration in cloud water and observations.*

**General comment#4:** It is unclear how ligand-promoted dissolution is limited under cloud conditions in the model. I would expect a more detailed discussion of the cloud parameters that influence oxalate production, such as liquid water content (LWC) and cloud cover, as well as how these factors are incorporated into the process of oxalate-promoted iron dissolution.

**Response**: Thanks for your insightful comment. Previous studies demonstrated that oxalate ligand complexation reaction proceeds within cloud environments. And the oxalate-promoted processing rates are measured in water used in our study from Paris et al. (2011). Hence, the oxalate-promoted processing only occurs in cloud environment set by our model as well as the study of Hamilton et al. (2019). Specifically, the cloud environment represented the cloud fraction is higher than 1E-5

and cloud liquid water content (LWC) is higher than 1E-8 (L/L(cloud)) in the model. The possibility of oxalate-promoted processing would increase if the cloud fraction and LWC increase. In the study area, we used the cloud fraction as an indicator to show the trend of cloud (Fig. 9e). Apart from the increased cloud faction, the temporal LWC was also increased and we added it as Figure S16. We have expanded the discussion in line 470 and line 493.

***Line 470:*** "Specifically, the cloud environment represents the cloud fraction is higher than 1E-5 and cloud liquid water content (LWC) is higher than 1E-8 (L/L(cloud)) in the model. The possibility of oxalate-promoted processing would increase if the cloud fraction and LWC increase. Hence, we focus our analysis on two factors including oxalate concentration and cloud fraction which is a proxy of cloud environment in this section."

***Line 494:*** "Similarly, the temporal LWC was also increased as shown in Figure S16. The increased cloud fractions and LWC would provide more possibility for oxalate-promoted processing as it only occurs in the cloud borne phase, thereby enhancing oxalate-promoted soluble iron production."

[Figure]

*Figure S16. (a) Spatial distributions of surface cloud liquid water content averaged of 2001-2017 springs. (b) Temporal variations of surface cloud liquid water content over high production rate area (30-45N, 120-150E) averaged of 2001-2017 springs.*


**General comment#5:** How much are the globally averaged dust emissions in the model? How is the emitted iron distributed between the fine (Aitken and accumulation) and coarse modes at dust emissions? What is the simulated global mean percentage of iron in dust? Additionally, what is the initial Fe solubility in dust? It would be beneficial for the reader to present some values used in the model, preferably in a separate table, instead of simply referring to the original publications.

**Response:** Thanks for your insightful comment. We have added the Table S3 which contained the global emissions of dust, dust total iron, dust soluble iron in fine and coarse mode, as well as the global mean iron content in dust and dust iron solubility. A discussion has been added in Section 3.2 in combination with General Comment #1.

*Line 272:*
"3.2 Spatial and temporal characteristics of dust iron deposition
3.2.1 Atmospheric dust iron budget
The global mean emissions of dust, dust total iron, and dust soluble iron were 2707 Tg/yr, 109 Tg/yr, and 0.98 Tg/yr, respectively, based on our 2017 model simulation (Table S3). These values are comparable to those reported by the Mechanism of Intermediate complexity for Modelling Iron (MIMI) model (Hamilton et al., 2019) (3200 Tg/yr and 130 Tg/yr for dust and dust total iron emissions) but are approximately twice as high as the results from EC-Earth model (Myriokefalitakis et al., 2022) (1265 Tg/yr and 59.3 Tg/yr). The simulated global mean iron content in dust is 4.0% which aligns well with MIMI (4.1%) and EC-Earth (4.7%). The initial iron solubility is 0.91% and higher than the 0.1% set by EC-Earth."

*Table S3. Global emissions of dust, dust total/soluble iron in 2017.*

|  | Dust | Total iron | Soluble iron | Iron content | Iron solubility |
|---|---|---|---|---|---|
| **Emissions (Tg/yr)** | 2707 | 109 | 0.98 | 4.0% | 0.91% |
| **coarse mode** | 2677 | 107 | 0.93 | 3.9% | 0.88% |
| **fine mode** | 30 | 2 | 0.05 | 5.3% | 3.4% |

**General comment#6:** The authors indicate that oxalate-promoted processing accounts for 25% of total soluble iron deposition from dust, approximately double that of proton-promoted processing. This finding is noteworthy, as it contradicts other studies suggesting an alternative perspective. For example, Ito and Shi (2016) reported that the proton-promoted dissolution scheme contributed the majority of soluble iron deposition to the ocean, while Myriokefalitakis et al. (2022) found that proton-promoted dissolution is the primary process for dust aerosols, whereas ligand-promoted dissolution is considered more significant for combustion aerosols (which are not addressed in this study). Could this outcome of the model indicate an underestimation of aerosol acidity or an overestimation of oxalate concentrations within the model? Is this result only attributable to coarse-mode dust? Could you please provide further elaboration on this finding?

**Response:** Thanks for your insightful comment. Below, we address the key aspects of the oxalate-promoted versus proton-promoted soluble iron contributions:

(1) Relative Contributions from Different Processes

The relative contributions of emissions, proton-promoted and oxalate-promoted to Northwest Pacific dust soluble iron deposition in coarse and fine modes are presented in Figure S8. Our findings highlight the dominant role of oxalate-promoted processing, particularly in the coarse mode as you mentioned. For the fine mode, proton-promoted processing accounts for approximately 39% of the soluble iron deposition which is about six times higher than oxalate-promoted processing. We have added Figure S8 and expanded the discussion in line 331 to further clarify these contributions.

(2) Consistency with Previous Studies

Globally and over East Asia, our study shows oxalate-promoted processing dominates atmospheric soluble iron deposition. This result aligns with the findings of Scanza et al. (2018), which demonstrated the significant role of oxalate-promoted processing using the CAM4 model. Additionally, observational data from the Qingdao station (Shi et al., 2022) further corroborate the critical role of oxalate in soluble iron deposition.

(3) Differences Between Models

Compared to the observations, the estimated oxalate in our model showed no significant over- or underestimation especially over the East Asia (Figure S8). Based on Figure 6g in Myriokefalitakis et al. (2022) and figure S3 in Ito (2015), oxalate concentrations appear to be underestimated. This suggests that the role of oxalate-promoted dissolution might also be underestimated in their model. In terms of proton-promoted processing, the simulated mainland coarse-mode aerosol acidity in our model (Figure S2) is obviously lower than that in EC-Earth (Fig. S3e; Myriokefalitakis et al., 2022) and IMPACT model (Fig. S2b; Ito and Xu, 2014). This could explain the reduced contribution of proton-promoted dissolution for dust soluble iron in our results compared to EC-Earth and IMAPACT. But The lack of sufficient observations makes it challenging to provide a detailed comparison for the model performance with respect to coarse mode aerosols. What's more, the acidity of fine aerosols simulated in our model have been validated by comparing with observationally estimated pH (Figure S2) showed no significant over- or underestimation.

(4) Model Limitations and Uncertainties

The higher contribution of oxalate-promoted dissolution in our study might be partially attributed to differences in the parameterization of oxalate concentrations and aerosol acidity between models. As discussed in General Comment #2, the limitations of our oxalate estimation method could also introduce uncertainties. To refine the relative contributions of ligand- and proton-promoted dissolution, we emphasize the importance of further observational constraints on oxalate concentrations, aerosol acidity, and their interactions in soluble iron production. Future studies should focus on improving these aspects to enhance model accuracy. We have made a further discussion in line 318.

*Line 318:* "Throughout the springs of 2001-2017, the NWP received an average of 4.9 Gg/season of soluble iron deposition from EA (Figure 5). The relative contributions of emissions, proton-promoted and oxalate-promoted to Northwest Pacific dust soluble iron deposition in coarse and fine modes are presented in Figure S8. Atmospheric processing played a significant role (~40%) in dust soluble iron deposition of which the oxalate ligand-promoted processing emerged as a dominant contributor (25%). The contribution of the oxalate-promoted processing was about twice that of

proton-promoted processing. And this result is consistent with previous modelling (Johnson and Meskhidze, 2013; Scanza et al., 2018) and observation research (Shi et al., 2022). Differently, Ito and Shi (2016) and Myriokefalitakis et al. (2022) found that proton-promoted dissolution is the primary process. The higher contribution of oxalate-promoted dissolution in our study might be partially attributed to differences in the parameterization of oxalate concentrations and aerosol acidity between models. As the oxalate concentrations appear to be underestimated in their model and the simulated mainland coarse-mode aerosol acidity in our model (Figure S2) is obviously lower than those. Future studies should focus on improving these aspects to refine the relative contributions of atmospheric processing."

*Line 331:* "And The dominant role of oxalate-promoted processing was mainly determined by the coarse mode (Figure S8). For the fine mode, proton-promoted processing accounts for approximately 39% of the soluble iron deposition which is about six times higher than oxalate-promoted processing."

[Figure]

*Figure S8. Relative contributions of emissions, oxalate-promoted, and proton-promoted processing to the Northwest Pacific dust soluble iron deposition averaged of 2001-2017 springs in total (a), coarse mode (b), and fine mode (c).*


**Response**: Thank you for your insightful comment. The evaluation of fine mode pH is based on observationally estimated ground-level fine-aerosol pH and annually averaged simulated values at consistent locations. The correlation coefficient and normalized mean bias (NMB) are 0.4 and 27% respectively. Regarding the potential reasons for the misrepresentation of atmospheric acidity in the model: We directly utilize the annually averaged simulated aerosol pH to compare with observations from different months. Hence, the seasonal variations may introduce discrepancies of approximately ±1 in the pH values. What's more, the dynamic changes of precursor gas emissions, environmental factors such as relative humidity could also induced discrepancy of modelled aerosol acidity.
We also have added figures of the calculated pH values for the accumulation mode and coarse mode aerosols as Figure S2 These figures clearly illustrate that fine particles are relatively more acidic while coarse-mode particles are significantly less acidic influenced by sea salt and dust components.

We have made a further discussion in line 158.

*Line 158:* "The simulated aerosol pH in accumulation and coarse mode have been shown in Figure S2. The fine particles are relatively more acidic while coarse-mode particles are significantly less acidic influenced by sea salt and dust components. Through the annually averaged comparison of accumulation mode aerosols' pH with observations collected by Pye et al. (2020), our model successfully captured the global characteristics of fine aerosol pH. The correlation coefficient and normalized mean bias (NMB) are 0.4 and 27% respectively. The discrepancy could be attributed by the seasonal variations and the dynamics of precursor gas emissions, environmental factors such as relative humidity."

[Figure]

*Figure S2. (a) Spatial distribution of aerosol pH in accumulation mode in 2013 and observationally estimated ground-level fine-aerosol pH (dots) from Pye et al. (2020). (b) The linear relationship between aerosol pH simulation and observationally estimated ground-level fine-aerosol pH. (c) Spatial distribution of aerosol pH in coarse mode in 2013.*

**General comment#8:** As a final comment, while the paper focuses on the deposition of soluble iron from dust aerosols, the omission of pyrogenic iron complicates a fair comparison with atmospheric observations. Numerous recent studies underscore the importance of pyrogenic iron from downwind source regions similar to the one examined here. It is unclear why the authors did not also include pyrogenic iron emissions in their analysis, especially since other versions of the model did. Consequently, when evaluating a model against observational data, the authors should preferably select cases where iron-containing dust aerosols predominantly influence the measured concentrations (e.g., by utilizing back trajectories). However, it is not clear whether this approach was implemented in the current study. Could you please provide some clarification on this?

**Response**: Thank you for your insightful comment. The primary goal of this study is to evaluate the trends in dust-derived soluble iron deposition which is most pronounced during the spring seasons when East Asian dust emissions significantly impact the Northwest Pacific. Pyrogenic iron emission shall not be a key factor to the analysis of long-term trend of natural dust iron, especially our analysis was made for spring only. For instance, the pyrogenic soluble iron deposition to the Northwest Pacific account for 36% of total soluble iron deposition from 1980 spring to 2014 spring (Hamilton et al., 2020), indicating dust shall be the dominant sources during spring. What's more, the uncertainties in current pyrogenic iron emission inventories (Ito et al., 2023; Rathod et al., 2020; Liu et al., 2024) make it challenging to incorporate this source accurately. Therefore, we did not include pyrogenic iron in our analysis as our focus remains on dust as the dominant contributor to soluble iron during the spring season. But the anthropogenic soluble iron deposition to the Northwest Pacific presented an obvious trend (increased ~1Gg/season) from 2001 spring to 2014 spring from the study of Hamilton et al. (2020). The increased trend could opposite the decreased dust soluble iron deposition to some extent. We have expanded the discussion in line 545.

The observations of iron used here is the total iron which include dust and pyrogenic iron. Our model only captures the 0-10% iron solubility (Figure S7). This might be due to the lack of

pyrogenic iron which has been suggested to contribute to higher iron solubility (Ito et al., 2019). To address the potential influence of mixed iron sources, we performed preliminary back-trajectory analyses for selected cases which presented three dust events during springs (Buck et al., 2013; Chen 2004). But the back-trajectory analysis cannot fully distinguish between dust and pyrogenic iron contributions, especially as air masses pass over regions like the North China Plain which may introduce anthropogenic influences in springs (showed below). We have clarified the limitation of observations and expanded the discussion in line 267.

*Line 545:* "It is crucial to acknowledge that this study focuses on spring dust sources of iron but pyrogenic iron sources, such as those from anthropogenic activities and biomass burning in other seasons also make a substantial contribution to the ocean's soluble iron inventory due to their high solubility (Ito et al., 2021; Ito et al., 2019; Rathod et al., 2020). The increased anthropogenic soluble iron deposition trend during our study period could opposite the decreased dust soluble iron deposition to some extent (Hamilton et al., 2020)."

*Line 267*: "The observations of iron used here is the total iron which include dust and pyrogenic iron. The simulated results are lower than observations likely due to the lack of pyrogenic iron. What's more, the comparison about iron solubility between simulation and observations has shown in Figure S7. Our model only captures the 0-10% iron solubility. This is likely due to the lack of pyrogenic iron which has been suggested to contribute to higher iron solubility (Ito et al., 2019)."

[Figure]

*Figure S7. The comparison about iron solubility between simulation and observations.*

[Figure]

*Figure. Examples of back-trajectory analyses*

**Response**: Thank you for your insightful comment. We have added the global distribution of initial iron content in dust coarse mode aerosol as Figure S1(a) and expanded the discussion in line 137.

*Line 138:* "According to the utilization of the mineralogy map, our model achieved to simulate the global spatial patterns of total and initial soluble iron emissions. Compared to the default setting of 3.5%, the total iron content in dust aerosol is higher in the main dust sources including North Africa, Middle East and central Asia, and East Asia (Fig. S1a). This is consistent with the observations (Lafon et al., 2004, 2006; Shi et al., 2011b) and the research by Ito and Xu (2014), which reported that the observed iron content in North Africa and East Asia averaged 3.7%. Therefore, the use of the mineralogy map increases the iron content in dust from these regions (Fig. S1b) which suggest the default settings likely underestimate dust iron in these main dust source regions."

[Figure]

*Figure S1. (a) Spatial distribution of iron content in coarse mode dust aerosol. (b) Compared to the default setting (3.5% iron in dust), changes in dust total iron surface concentrations from the developed model averaged 2001-2017 springs.*

(2) Lines 252-254: Can you please also provide actual rates rather than just percentages? How are these numbers compared to other studies?

**Response**: Thank you for your insightful comment. Combined with the General comment#1, we have added the annually mean dust total/soluble iron deposition in the Northwest Pacific in Table 2 and compared with other studies in line 287.

*Line 287:*
"3.2 Spatial and temporal characteristics of dust iron deposition
3.2.1 Atmospheric dust iron budget
…For the Northwest Pacific, the simulated dust total and soluble iron burdens are 7.4 Gg and 0.11 Gg respectively. These values align with those reported by the Ensemble model. The simulated dust total and soluble iron deposition rates in the NWP are 589 Gg/yr and 10.1 Gg/yr which are consistent with results from both MIMI and the Ensemble model. But the NWP dust iron deposition reported by EC-Earth is higher than in our study due to differences in regional dust emissions."

(3) Section 3.2: Deposition rates could also be presented in a table.

**Response**: Thank you for your insightful comment. Combined with the General comment#1, we have added the annually mean dust total/soluble iron deposition in Table 2 in section 3.2.1.

*Table 2. Global annual atmospheric dust total/soluble iron budget in 2017.*

|  | Burden (Gg) | | Dry deposition (Tg/yr and Gg/yr) | | Wet deposition (Tg/yr and Gg/yr) | | Solu. Rate (Gg/yr) | |
| --- | --- | --- | --- | --- | --- | --- | --- | --- |
|  | Fe_tot | Fe_sol | Fe_tot | Fe_sol | Fe_tot | Fe_sol | Fe_ps | Fe_os |
| **Global** | 1331 | 15.4 | 42.7 | 445 | 66.5 | 874 | 129 | 200 |
| **coarse, fine** | 1301, 30 | 13.3, 2.1 | 42.2, 0.5 | 421, 24 | 65.4, 1.1 | 789, 85 | 78, 51 | 196, 4 |
| **NWP** | 7.4 | 0.11 | 0.05 | 1.2 | 0.54 | 8.9 | 1.3 | 1.73 |
| **coarse, fine** | 7.1, 0.3 | 0.09, 0.02 | 0.05, 0.001 | 1.1, 0.1 | 0.53, 0.01 | 7.8, 1.1 | 1.1, 0.2 | 1.68, 0.05 |

(4) Table 1: In general, all emissions come from CMIP6 with MEIC for China. Does this information really need to be repeated in the table? Moreover, why are chlorine emissions not presented in the table?

**Response**: Thank you for your insightful comment. We have removed the repeated words in the table. And the gas HCl is the model is not originated from primary emissions but secondary source.

(5) Figure 3: I propose to color-code only the seasons, not the 12 months of the year. It would probably make the figure less noisy.

**Response**: Thank you for your insightful comment. We have changed the color setting as follows.

[Figure]

(6) In the whole manuscript: Better to change "oxalate-ligand-promoted" to either oxalate- or ligand-. Usually oxalate is used as a proxy for all organic ligands.

**Response**: Thank you for your insightful comment. We have replaced the "oxalate-ligand-promoted" to "oxalate-promoted".

(7) Line 343: The NCP abbreviation needs to be explained.

**Response**: Thank you for your insightful comment. We have explained the NCP in line 395.

*Line 395:* "… especially over North China Plain with intensive $NO_x$ emission (Luo et al., 2020b)."

(8) Line 344: Please explain why HCl concentrations are increased in the model. How have precursor emissions changed?

**Response**: The gas HCl is the model is not originated from initial emissions but secondary source. In MOSAIC, the HCl gas are from irreversible heterogeneous reactions between acidic gases (such as $HNO_3$) and salt of chloride (NaCl and $CaCl_2$). It is the increased $NO_x$ emissions induced higher $HNO_3$ gas and then more HCL are product. The increased $HNO_3$ gas has been shown in Figure S7. We have expanded the discussion in line 424 and corrected the earlier term of HCl emissions in line 458.

*Line 424:* "And the increasing trend in HCl concentration was induced by the enhanced $HNO_3$ gas which would produce HCl gases through heterogeneous reactions with $NaCl/CaCl_2$."

*Line 458:* "On the one hand, the increase in iron solubility can be attributed to enhanced $NO_x/HCl$ concentrations and reduced dust emissions"

[Figure]

*Figure S7. Temporal variations of surface concentrations of $SO_2$ (a), NOx (a), HCl (c), and $HNO_3$(d) over the high production rate of proton-promoted soluble iron area (30-45N, 120-150E) averaged of 2001-2017 springs.*

(9) Lines 366-367: Can you please explain why aerosol water content is increased due to enhanced $NO_x$? How much has the coarse nitrate changed during the studied period? Please also discuss this, taking into account the general comments.

Response: Thanks for your insightful comment. We have corrected the mistake of 'coarse mode aerosol' to 'fine mode aerosol' in line 447. The decrease fine mode aerosol acidity over the mainland of EA could be attributed to the increased aerosol water content. This is caused by enhanced $NO_x$. Due to the high hygroscopicity of nitrate aerosol, the increased $NO_x$ emissions would induce higher nitrate aerosol content and aerosol water content.
Specifically, aerosol water content in MOSAIC is calculated used Zdanovskii-Stokes-Robinson

(ZSR) mixing rule as the functions shown below:

$$W = \sum_{E=1}^{N} \frac{n_E}{m_E^0(a_w)}$$

where W is the aerosol water content, $n_E$ is the number of moles of any electrolyte E in the solution, $m_E^0(a_w)$ is the binary electrolyte molality of E at the solution water activity which assumed $a_w$=RH. The increased nitrate content of high hygroscopicity would results in higher water content. The changes of fine nitrate aerosol and water content due to the increase $NO_x$ emission experiment has shown in Figure S14.

We have corrected the mistake in line 447 and added the figure S14 to illustrate.

*Line 366:* "The decreased fine mode aerosol acidity over the mainland of EA could be attributed to the increased aerosol water content induced by enhanced $NO_x$ (Figure S14).

[Figure]

*Figure S14. Spatial distributions of surface accumulation mode nitrate aerosol concentration (a) and aerosol water content (b) induced by $NO_x$ experiment.*

---

## Author Response (AR2)

Dear authors, thank you for the careful revision of your manuscript.

Please perform the following corrections (lines refer to the track changes document)

**Summary Response**: Thank you for your careful review of the manuscript. We have addressed all the comments and suggestions as outlined below. Please find our detailed responses and corresponding revisions (lines refer to the track changes document).

(1) - Line 110: "mol/m-3 (air)" please replace "/" by "." or empty space

**Response**: Thank you for your suggestion. We have revised "mol/m$^{-3}$ (air)" to "mol m$^{-3}$ (air)" in line 110 as recommended.

(2) - Lines 112-114: please clarify in the text whether the concentrations used for pH calculations are monthly mean values or something else.

**Response**: Thank you for your suggestion. We have clarified the methodology for pH calculations as follows:

Line 114: "The aerosol pH is calculated based on H$^+$ concentrations for each aerosol mode at each time step."

(3) - Lines 137-138: please replace "According to the utilization of" by "Using"

**Response**: Thank you for your suggestion. We have replaced "According to the utilization of" with "Using" in line 138.

(4) - Lines 139-140: please replace the sentence starting by "Compared to the default setting of 3.5%, ...." by "The total iron content in dust aerosol in our model in the main dust sources, including North Africa, Middle East and Central Asia, is higher than the default setting of 3.5% (Fig. S1a)" and modify accordingly figure S1 caption.

**Response**: Thank you for your suggestion. We have revised the sentence in the manuscript and supplement as follows:

Line 140: "The total iron content in dust aerosol in our model in the main dust sources, including North Africa, Middle East and Central Asia, is higher than the default setting of 3.5% (Fig. S1a)."

Line 14 in the supplement: "Changes in dust total iron surface concentrations from the developed model compared to the setting of 3.5%, averaged over the 2001-2017 springs."

(5) - Line 142: the observed DUST iron content

**Response**: Thank you for your suggestion. We have added "dust" to clarify the context. The revised phrase now reads:

Line 143: "the observed dust iron content."

(6) - Line 144: suggest THAT

**Response**: Thank you for your suggestion. We have replaced "which" with "that" in

line 144 as recommended.

(7) - Line 154: is this eq 5a you refer to?

**Response**: Thank you for your suggestion. We have corrected the reference to "Eq. 5a" instead of "Eq. 5" in line 155.

(8) - Line 160: replace "have been shown" by "are shown"

**Response**: Thank you for your suggestion. We have replaced "have been shown" by "are shown" in line 161.

(9) - Line 161: replace "Through the annually averaged comparison of" by "The comparison of annually averaged..."

**Response**: Thank you for your suggestion. We have restructured the sentence as follows:

Line 162: "The comparison of annually averaged accumulation mode aerosols' pH with observations collected by Pye et al. (2020) shows that…"

(10) - Line 162: add after the reference "that"

**Response**: Thank you for your suggestion. We have restructured the sentence as follows:

Line 162: "The comparison of annually averaged accumulation mode aerosols" pH with observations collected by Pye et al. (2020) shows that…"

(11) - Line 164: please replace "by" by "to"

**Response**: Thank you for your suggestion. We have replaced "by" by "to" in line 165 as recommended.

(12) - Lines 181,182: please specify if it is upper or lower limit threshold

**Response**: Thank you for your suggestion. We have specified the upper threshold in line 181.

(13) - Lines 185,186: "The locations and months are consistent between observations and the model." Are monthly mean modelled values are used? What about the year of the observations is the same with the simulated values or you consider this as climatological mean? Please describe precisely what you have done.

**Response**: Thank you for your suggestion. We have clarified that monthly mean modelled values were used, and the observations were treated as climatological means over the period 2001-2017. The revised text reads:

Line 186: "The locations and months are consistent between observations and the model. We utilized monthly mean modelled values, averaged climatologically over the period 2001-2017."

(14) - Line 272- 274: The observations of iron used here ARE the total iron which INCLUDES dust ... between simulation and observations IS shown ...Our model only CALCULATES IRON SOLUBILITIES BETWEEN 0 and 10%

**Response**: Thank you for your suggestion. We have revised the sentence as follows:

Line 271: The observations of iron used here are the total iron which includes dust and pyrogenic iron. … What's more, the comparison about iron solubility between simulation and observations is shown in Figure S7. Our model only calculates iron solubility between 0 and 10%.

(15) - Lines 328 - 332: here it seems that you compare regional (NWP) with global budgets. Please clarify. This is particularly important because your model also performs well for OXL concentrations over East Asia but not elsewhere as shown in your figure S4b.

**Response**: Thank you for your suggestion. The primary role of oxalate is illustrated by our model both in the NWP and globally (Table 2 and line 287-293). We have clarified the comparison as follows:

Line 324: "The contribution of the oxalate-promoted processing was about twice that of proton-promoted processing in the NWP. This finding is consistent with global results (Table 2) and aligns with previous global modelling (Johnson and Meskhidze, 2013; Scanza et al., 2018) and East Asian observational research (Shi et al., 2022). Differently, …. As the oxalate concentrations appear to be underestimated in their model and the simulated mainland coarse-mode aerosol acidity in our model (Figure S2) is obviously lower than those. Furthermore, regional differences can also play a role, as our model performs well for oxalate concentrations over East Asia but not elsewhere (Figure S4). Future studies should …"

(16) - Line 347-348: Please specify to which period the rate corresponds. Is this a decrease over the studied period (then it is not a rate).

**Response**: Thank you for your suggestion. It is the variation in 2017 compared to 2001 spring. We have revised the sentence as follows:

Line 345: "However, the amount of soluble iron deposition produced from atmospheric processing showed a much lower decrease (18%) in 2017 compared to 2001 spring (Figure S9). The coarse-mode proton-promoted soluble iron deposition even increased by 7% as shown in Fig. S9d."

(17) - In the supplement, please change "oxalate-ligand" to "oxalate" everywhere

**Response**: Thank you for your suggestion. We have changed "oxalate-ligand" to "oxalate" throughout the supplement.

(18) - Figure S2: explicitly write in the caption that 1) the spatial distribution shown is near-surface, 2) the relationship shown in (b) is for fine aerosol, 3) how the aerosol pH was calculated (using monthly mean concentrations ?)

**Response**: Thank you for your suggestion. We have updated the caption to explicitly state the details as follows:

Line 18 in supplement: "Figure S2. (a) Spatial distribution of surface aerosol pH in

accumulation mode in 2013 and observationally estimated ground-level fine-aerosol pH (dots) from Pye et al. (2020). (b) The linear relationship between simulated surface aerosol pH in accumulation mode and observationally estimated ground-level fine-aerosol pH. (c) Spatial distribution of surface aerosol pH in coarse mode in 2013. The aerosol pH is calculated based on $H^+$ concentrations for each aerosol mode at each time step."

(19) - Explain colored symbols in Figure S4 caption

**Response**: Thank you for your suggestion. We have added an explanation of the colored symbols.

Line 36 in the supplement: "(b) The comparison between estimated oxalate concentration in cloud water and observations. Red circles represent locations in East Asia (EA), and green circles represent locations elsewhere."

(20) - Figure S10 caption: Interannual variation of the percent contribution of each iron solubilization process to the total soluble iron deposition

**Response**: Thank you for your suggestion. We have revised the caption as follows:

Line 63 in the supplement: "Figure S9. Interannual variations of dust soluble iron deposition from proton-promoted (a, d), oxalate-promoted (b, e) and emissions (c, f) in coarse and fine mode (atiken + accumulation) to the Northwest Pacific averaged of 2001-2017 springs."

Line 68 in the supplement: "Figure S10. Interannual variations of contribution of each iron solubilization process including proton-promoted (a, d), oxalate-promoted (b, e) and emissions (c, f) in coarse and fine mode (atiken + accumulation) to the dust total iron deposition to the Northwest Pacific averaged of 2001-2017 springs."

(21) - Figure S13 caption: "high production rate of ... area (...)" do you mean "area of high contribution of..."

**Response**: Thank you for your suggestion. We have updated the sentence as follows:

Line 83: "over the area of high production rate of proton-promoted soluble iron"

(22) - Figure S14 caption: CHANGES IN THE spatial distribution ....

**Response**: Thank you for your suggestion. We have added the "Changes in the" to the caption in line 87.

---

## Author Response (AR3)

Dear authors, thank you very much for the careful corrections. Please correct in the supplement in the captions of figures S9 and S10 the spelling of 'aitken'

**Response**: Thank you for your suggestion. We have revised "atiken" to "aitken" in the supplement as recommended.